# Co-evolution of tumor and immune cells during progression of multiple myeloma

Ruiyang Liu[1,2,7], Qingsong Gao[1,2,7], Steven M. Foltz[1,2,7], Jared S. Fowles[1], Lijun Yao[1,2],
Julia Tianjiao Wang[1,2], Song Cao[1,2], Hua Sun[1,2], Michael C. Wendl[2,3,4], Sunantha Sethuraman[1,2],
Amila Weerasinghe[1,2], Michael P. Rettig[1], Erik P. Storrs[1,2], Christopher J. Yoon[1,2],
Matthew A. Wyczalkowski[1,2], Joshua F. McMichael[1,2], Daniel R. Kohnen[1], Justin King[1], Scott R. Goldsmith[1],
Julie O'Neal[1], Robert S. Fulton[2], Catrina C. Fronick[2], Timothy J. Ley[1], Reyka G. Jayasinghe[1,2], Mark A. Fiala[1],
Stephen T. Oh[1,5], John F. DiPersio[1,6], Ravi Vij[1,6✉] & Li Ding[1,2,3,6✉]

Multiple myeloma (MM) is characterized by the uncontrolled proliferation of plasma cells. Despite recent treatment advances, it is still incurable as disease progression is not fully understood. To investigate MM and its immune environment, we apply single cell RNA and linked-read whole genome sequencing to profile 29 longitudinal samples at different disease stages from 14 patients. Here, we collect 17,267 plasma cells and 57,719 immune cells, discovering patient-specific plasma cell profiles and immune cell expression changes. Patients with the same genetic alterations tend to have both plasma cells and immune cells clustered together. By integrating bulk genomics and single cell mapping, we track plasma cell subpopulations across disease stages and find three patterns: stability (from precancer to diagnosis), and gain or loss (from diagnosis to relapse). In multiple patients, we detect "B cell-featured" plasma cell subpopulations that cluster closely with B cells, implicating their cell of origin. We validate AP-1 complex differential expression (JUN and FOS) in plasma cell subpopulations using CyTOF-based protein assays, and integrated analysis of single-cell RNA and CyTOF data reveals AP-1 downstream targets (IL6 and IL1B) potentially leading to inflammation regulation. Our work represents a longitudinal investigation for tumor and microenvironment during MM progression and paves the way for expanding treatment options.

[1] Department of Medicine, Washington University in St. Louis, St. Louis, MO, USA. [2] McDonnell Genome Institute, Washington University in St. Louis, St. Louis, MO, USA. [3] Department of Genetics, Washington University in St. Louis, St. Louis, MO, USA. [4] Department of Mathematics, Washington University in St. Louis, St. Louis, MO, USA. [5] Center for Human Immunology and Immunotherapy Programs, Washington University in St. Louis, St. Louis, MO, USA. [6] Siteman Cancer Center, Washington University in St. Louis, St. Louis, MO, USA. [7] These authors contributed equally: Ruiyang Liu, Qingsong Gao, Steven M. Foltz. ✉email: rvij@wustl.edu; lding@wustl.edu

**M**ultiple myeloma (MM) is a disease characterized by clonal proliferation of malignant plasma cells, sometimes manifesting clinically with anemia, renal impairment, and pathologic bone fractures[1,2]. Over the past three decades, novel therapies, such as autologous hematopoietic cell transplantation, proteasome inhibitors, immunomodulatory drugs, and targeted monoclonal antibodies have led to dramatic improvements in quality and length of life in patients with MM[3-7]. Despite these advances, the disease remains incurable for most patients as it progresses and becomes resistant to these treatments.

Several landmark genomic studies have led to a greater understanding of the molecular pathogenesis of myeloma. These studies have demonstrated recurrent mutations in *KRAS, NRAS,* and *TP53* as well as a significant percentage of previously unrecognized mutations affecting RNA processing and protein homeostasis[8-10]. Other investigations have used bulk sequencing technologies to broadly describe MM clonal heterogeneity and evolution in terms of shifting subclonal dominance and branching evolution, often in response to therapeutic selective pressure[7,11,12]. There is an impetus to translate the growing understanding of the genomic landscape of MM into precision therapies. This is highlighted by the upcoming MyDRUG trial (NCT02884102) being initiated by the Multiple Myeloma Research Foundation (MMRF), which will use genomic and transcriptomic information obtained from the CoMMpass study (relating clinical outcomes to assessment of individual genetic profiles) in order to identify targetable genetic alterations and to evaluate personalized therapies to enrollees.

Single-cell sequencing methods combine novel sequencing technologies with cell-sorting techniques, allowing for a more granular understanding of inter- and intra-tumoral genomics[13]. Early studies used low-throughput systems to analyze the tumor microenvironment in solid tumors, examining the genomes and transcriptomes of malignant cells, as well the immune compartment, confirming the importance of single-cell resolution[13-15]. With the advent of high-throughput methods, these technologies are rapidly expanding toward dissecting all malignancies. Ledergor et al.[16] recently used single-cell RNA sequencing (scRNA-seq) to compare plasma cell transcriptomes from patients with newly diagnosed MM (NDMM), precursor states, and healthy controls; they highlighted significant inter-individual heterogeneity and demonstrated variable subclonal divergence leading to new thoughts about the role of intergenic mutations, epigenetics, and environmental transcriptional regulation. Jang et al.[17] used scRNA-seq to examine 597 CD138+ plasma cells from 15 patients at different stages of MM, associating clusters of gene expression with risk of early disease progression and cytogenetic abnormalities. The aforementioned studies, however, did not examine MM patients at multiple points during their disease progression, nor did they evaluate dynamic alterations in non-malignant components of the tumor microenvironment.

Here, we report our analysis of single-cell patterns in 29 longitudinal samples procured at different disease stages from 14 MM patients. We collectively analyzed 74,386 single cells from these patients, including 17,267 plasma cells and 57,719 immune cells. Deeper dissection of plasma cells and B cells identified subpopulations of plasma cells with various genetic changes and marker gene expressions, suggesting cells in transitional states. By single-cell sequencing, we discerned co-evolution maps of tumor and immune cells between smoldering MM (SMM) and primary stages and between primary and relapse stages after remission. In summary, our study represents a longitudinal investigation of tumor and immune microenvironment during MM disease development and paves the way for expanding treatment options for this disease.

## Results

### Patients, treatments, technologies, and landscape of genomic alterations in MM.
The main data corpus of the study comprises 29 longitudinal samples from 14 individuals with different combinations of disease stages, sequencing data types, and treatments (Fig. 1a, Supplementary Fig. 1a and Supplementary Data 1). All patients have at least one sample with both single-cell RNA sequencing (scRNA-seq) and 10x Genomics linked-read whole-genome sequencing (10xWGS), and nine patients have data from two or more time points, including a mix of CD138+ sorted and unsorted bone marrow aspirate samples. Three patients have data from the SMM and primary stages, and six have both primary and relapse samples. To ensure samples matched across time points, we compared germline variant allele fractions (VAFs) at 24 loci (Supplementary Fig. 1b and Supplementary Data 1). In addition, we performed CyTOF-based profiling and validation using tumor samples from four additional patients.

MM exhibits a variety of primary and secondary genomic events (Fig. 1b and Supplementary Figs. 1c and 2d). We analyzed potential driver events, focusing on known significantly mutated genes and structural and copy number variation (CNV) (Fig. 1b and Supplementary Data 1). Three patients had hyperdiploid (HRD) copy number profiles with little evidence of translocation events, and in 12 patients, we observed loss of 13q supported by at least one level of evidence[18,19]. Most translocations in MM involve the highly expressed *IGH* locus on chromosome 14, with t (11;14) being the most frequent[20] and t(4;14) being associated with adverse prognosis[21-25]. We have multiple evidence levels of t (11;14) in three patients and t(4;14) in one patient.

We detected a median of 55 coding mutations from whole-exome sequencing (WES) and 6702 total mutations from whole-genome sequencing (WGS) (Supplementary Data 1). The VAF distribution was consistent across sequencing platforms for key driver mutations, including *TP53, NRAS, KRAS,* and *DIS3* (refs. [26-28]). We observed VAF changes during disease progression for several mutations in cancer genes, notably *TP53* and *NRAS* in Patient 27522. Specifically, *TP53*-R248Q expands from 0.4% to 33.1%, while *NRAS*-Q61K recedes from 17.1% to 0.6% during progression from Primary to Relapse-1 (Supplementary Fig. 1c and Supplementary Data 1).

### Tumor and immune populations influenced by genetic alterations and treatments during disease progression.
We integrated scRNA-seq data from all 14 patients; after quality control and cell type detection ("Methods"), we retained 74,986 cells from 11 patients, including 17,267 plasma cells and 57,719 non-plasma cells. The proportions of plasma and immune cell types vary across patients and disease stages (Fig. 2a). Plasma cells in primary tumor samples ranged from 0.9% to 84.1%. Other cell types detected include B cells (3686), macrophages (16,183), monocytes (4249), CD4+ T cells (18,250), CD8+ cells (8334), natural killer cells (6282), and dendritic (DC) cells (735) (Fig. 2a and Supplementary Fig. 3B). Different patients show a range of cell type compositions, such as complete loss of NK cells in Patient 27522 at the primary stage, but presence of 22% NK cells in Patient 77570 at the primary stage. Different stages from the same patient can also have different compositions as well. For example, in Patient 59114, CD4+ T cells change from 36% at Primary to 9% at both pre- and post-transplant, and increase back to 35% at Relapse-1.

Mapping somatic mutations to individual scRNA cells has the potential to identify tumor cells that cannot be discerned purely by expression data or subclonal populations with different mutational patterns[29]. Overall, we mapped 48 mutations to 198 cells from 14 samples (Supplementary Fig. 2b, c, Supplementary Data 2 and Methods). Variants in key driver genes, such as *NRAS*

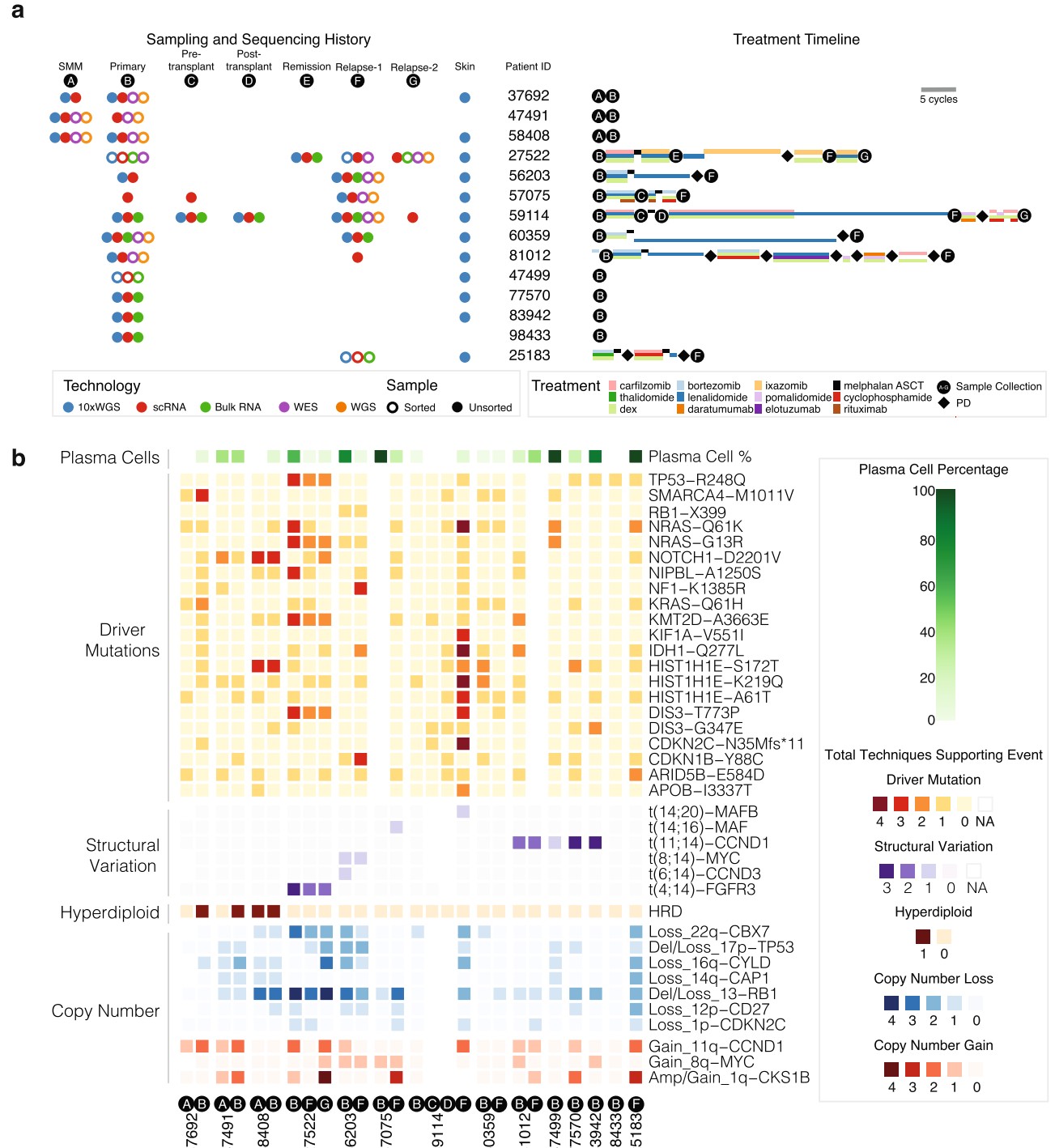

**Fig. 1 Samples, next-generation dataset, and genomics landscape. a** Sample type, technology, and treatment timeline broken down by patient. Left portion shows sample technology (10xWGS, scRNA, Bulk RNA, WES, WGS) and sample type (CD138+ sorted vs. unsorted). Right portion shows each patient's treatment timeline. Treatment length corresponds to the number of cycles. SMM smoldering multiple myeloma. **b** Heatmap shows the landscape of copy number variations (CNV), structural variants (SV), and driver mutations across 14 patients. Copy number amplification/gain, copy number deletion/loss, SV, and driver mutations are shown in red, blue, purple, and orange, respectively, with colors indicating the number of techniques supporting the event. Techniques for copy number events are FISH, 10xWGS, regular WGS, WES, and scRNA-seq. Techniques for SV are FISH, 10xWGS, Bulk RNA-seq, and scRNA-seq. Techniques for driver mutations are 10xWGS, WES, WGS, and Bulk RNA-seq. Number of techniques supporting an event is 0 if the only technique supporting the event is from scRNA-seq. Plasma cells percentage inferred from scRNA-seq is shown on the top of the heatmap.

G13R mutation, were primarily detected in plasma cells (158 cells) relative to non-malignant cell types (39 cells), which are much more numerous. The reference allele was detected more readily across cell types (1212 plasma cells and 5278 non-plasma cells) (Fig. 2b). We also examined mutations co-residing in the

same cells (Supplementary Fig. 2a), finding co-existing mutations such as NRAS-G13R with *YBX1*-F74L/*ACAT1*-S14N/*CLPTM1L*-T33S in the 27522 Relapse-2 sample.

Single-cell expression profiles of plasma cells primarily clustered by each individual patient, with different disease stages

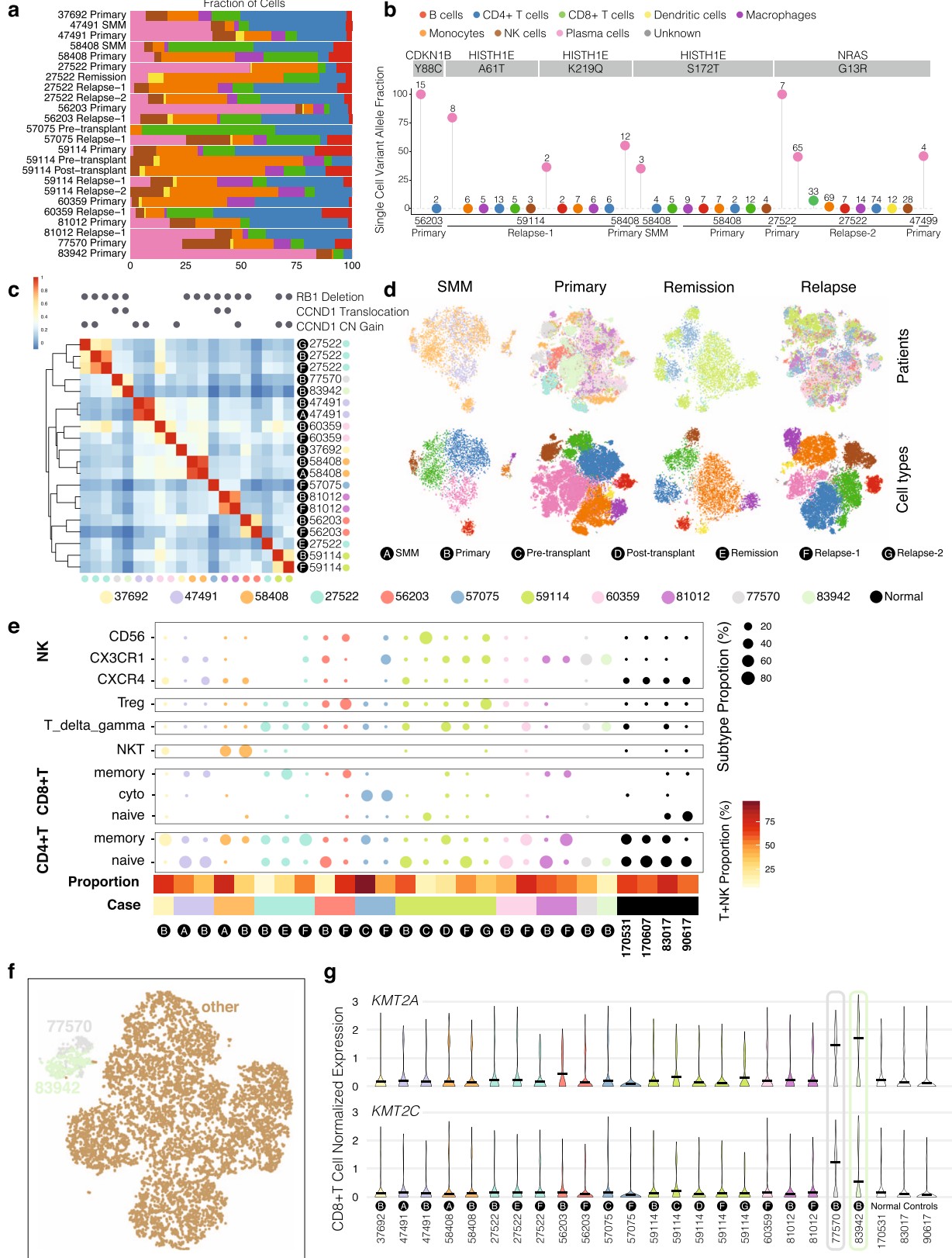

of the same patient showing high similarity (Fig. 2c and Supplementary Fig. 4a, b), while expression of non-plasma cells largely cluster by cell types (Supplementary Fig. 3a). Notably, we observed the highest correlation between SMM and primary tumors (0.92 for Patient 47491 and 0.91 for Patient 58408), but lower and more variable correlation between primary and relapse samples in other patients, which could reflect treatment as a factor in altering expression profiles of malignant plasma cells. Expression profiles also partially clustered by genetic alterations; Patients 77570 and 83942 both harbor *CCND1* translocation, and their plasma cell expression profiles are more similar than others (Fig. 2c, Supplementary Data 3 and Methods).

**Fig. 2 Integration analysis across 14 multiple myeloma patients revealing distinct cancer populations and immune microenvironments during disease progression. a** Bar plots showing cell type fractions for each sample. Colors indicate cell type. **b** Single-cell variant allele fractions (VAF) for driver mutations. Each bubble is colored by the cell type with the associated VAF, and total cells supporting the variant are labeled atop each bubble. **c** Heatmap showing pairwise correlation of average expression for malignant cells in each sample. Genomic alterations with either FISH evidence or at least another two levels of evidence shown above. **d** t-SNE plots showing the integration of samples from multiple patients for a given time point. Clustering of cells from different time points are colored by patient (top) or by cell type (bottom). The remission group includes one remission sample, one pre-transplant, and one post-transplant. **e** Proportion of T/NK cells within the total T/NK cell cohort for each sample. The proportion of T and NK cells within each sample is shown at the bottom as a color bar. **f** t-SNE plot showing CD8+ T cells from all the patients where CD8+ T cells are available. Cells from the primary sample of Patients 77570 and 83942 and Relapse-2 sample of Patient 27522 are colored specifically. **g** Expression pattern of *KMT2A* and *KMT2C* in CD8+ T cells for each sample.

We also integrated samples from multiple patients by disease stage (Fig. 2d, top row colored by patient, bottom row colored by cell type). We observed again that plasma cells tended to cluster by patient, and found that non-plasma cells clustered by cell type and included a broader mix of patients. We then identified genes with variable expression across disease stages in multiple patients. For example, we found CD4+ T cells from primary tumors show a higher expression of *NFKBIA* when compared to SMM. In Patient 27522, *NFKBIA* expression was lost during remission, but regained in relapse. *NFKBIA* is a negative regulator of NF-kB, meaning cell types with higher *NFKBIA* might implicate altered NF-kB activity. In another example, we found higher expression of *CD69* in CD4+ T cells of remission samples, which was subsequently lost during relapse. Higher expression of *IL1R2* was observed in primary sample monocytes but was then lost in remission monocytes. In monocytes found in the Relapse-2 sample of Patient 59114, there was a slight increase in *IL1R2* expression, and a similar trend was observed for *IL1B* expression in the monocytes of Patient 60359 (Supplementary Fig. 3d). Together these suggest a role of *IL1* signaling during myeloma, which should be further explored.

To evaluate differences of the tumor microenvironment across patients, we did another integration including an additional four samples from healthy donors. Relapse-2 of case 27522 was excluded because it is the only sample with 5′ sequencing within the cohort. We then extracted cells from each non-tumor population for subclustering analysis. Within T cells, we identified naïve, cytotoxic, memory, and regulatory T cells, as well as NKT and T delta gamma cells based on typical marker expression[30] (Fig. 2e). For NK cells, we also recapitulated three populations: CXCR4+, CX3CR1+ and CD56+. While we were not able to find a common trend for how the tumor microenvironment evolves with disease progression, there are interesting cases worth noting. For case 56203, we noted the presence of regulatory T cells (Tregs) from both primary and relapse time points, with the primary time point having a higher proportion of Tregs. In case 59114, we observed the Treg population in the primary time point. This population was observed at very low levels in remission and first relapse, but later re-emerged at the second relapse. Meanwhile, for NK populations, the primary time point is dominated by the CXCR4+ population, which is replaced by CD56+ population during the pre-transplant stage. After transplant, the CX3CR1+ NK population becomes dominant, and the immune profile remains stable until the first relapse. However, in the second relapse stage, we found the CX3CR1+ population decreases with the re-emergence of the CD56+ NK population (Fig. 2e). These observations show a highly dynamic microenvironment profile over this patient's disease course. In case 58408, we observed the high presence of NKT and CD4+ memory T cells in both SMM and primary stages, suggesting a parallel evolution model within the microenvironment.

We also observed that cells from some samples exhibited a consistent outlier pattern across cell types. This phenomenon is

particularly seen for the Primary samples of Patients 77570 and 83942, two cases with t(11;14) translocation. Specifically, NK cells and especially CD4+ and CD8+ T cells from 77570 and 83942 overlapped, showing similar overall expression profiles and further suggesting their shared genetic alterations could shape similar tumor microenvironment (Fig. 2f and Supplementary Fig. 3b). For example, both cases are highly enriched in CX3CR1+ NK cells (Fig. 2e); in CD8+ T cells of these three samples, there is a higher expression of *KMT2A* and *KMT2C*, two genes belonging to the lysine methyltransferase family, suggesting epigenetic changes in the T cell population (Fig. 2f, g). There is strong evidence of t(11;14) (*CCND1* translocation) in Patients 77570 and 83942, suggesting further study into the role these events might play in modifying the tumor microenvironment. We found high expression of *CTSS* in the macrophages and monocytes of Patient 77570 (Supplementary Fig. 3c). *CTSS* encodes Cathepsin S, a major endoprotease processing the MHCII complex prior to antigen presentation. It has been shown in mouse models that CTSS is necessary for the release of IL1B in macrophages[31], and that macrophage-derived cathepsin S induces chemoresistance in breast cancer[32] and invasion in pancreatic cancer[33]; this could implicate the complex interaction within the tumor microenvironment.

**Delineating B cell lineage by gene signature analysis and genetic alteration mapping.** To study B cell lineage and the transition between normal and malignant plasma cells, we integrated B cells and plasma cells from 21 tumor samples with both cell types along with four healthy donors ("Methods"). After integration, we found that clusters separated by cell type (Fig. 3a), with mature B cells from each patient mapping to the same cluster as B cells from normal samples. There are three small B cell clusters (Fig. 3a, b), predominantly from healthy donors, that exhibit high expression of *SOX4*, *VPREB3*, and *MME*, suggesting a primitive B cell state[34,35]. Interestingly, we found plasma cells from healthy donors mixing with some MM plasma cells, suggesting that these particular MM plasma cells exhibit an expression pattern similar to normal plasma cells. The rest of the MM plasma cells largely clustered by patient, as shown previously (Supplementary Fig 4a).

To investigate whether the malignancy of plasma cells is implicated from the early B cell stages, we also subset only the B cell populations for analysis. We found cells from some patient samples along with two normal samples (090617 and 170531) to be outliers. We found substantial B cell signatures in the plasma cells—with high expression of typical plasma cell markers, such as *SDC1* and *TNFRSF17*—for Patients 56203 and 83942, and to a lesser degree for 77570, as illustrated by expression of B cell marker *MS4A1* (Supplementary Fig. 4c). It has been previously reported that a subset of patients with high *CCND1* expression exhibits a B cell phenotype (CD2 group)[22], consistent with our observation for 77570 and 83942. Patient 81012, harboring a *CCND1* translocation, had elevated expression of *FYN* and

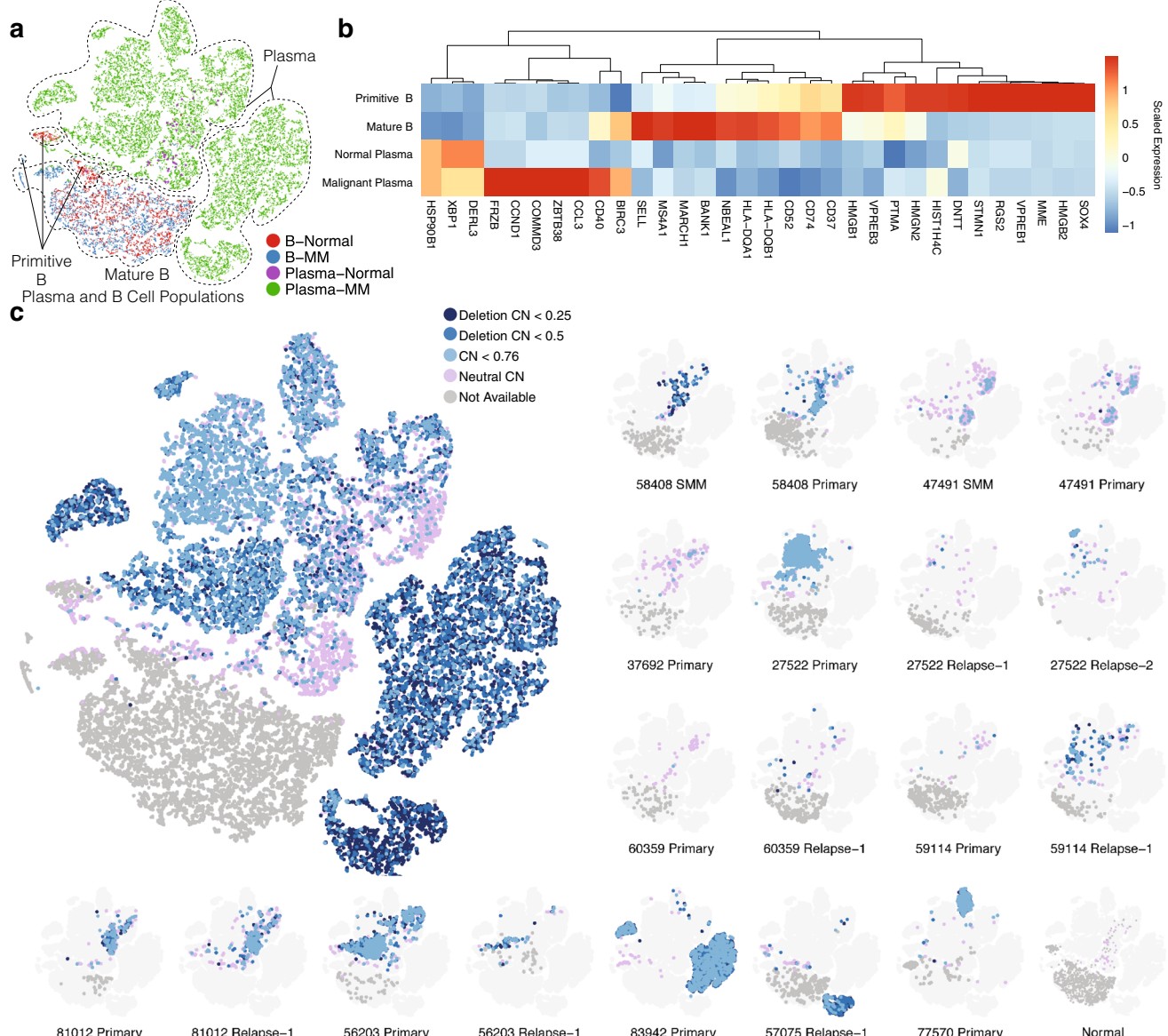

**Fig. 3 Analysis of B cell lineage markers and landscape for copy number events. a** t-SNE plot showing the distribution of B cells and plasma cells from all patients and four healthy "normal" donors. Dots are colored by samples of origin (patients/healthy donors) and cell type. **b** Heatmap showing genes specifically expressed at certain stages of B cell development. **c** Landscape of chromosome 13 deletion status showing all samples (left), with sample-specific maps for samples with at least one cell with chromosome 13 copy number (CN) < 0.76 (right). Within each t-SNE plot, dots are colored by chromosome 13 deletion status predicted from inferCNV. Dots colored in grey are B cells with CNV scores unavailable.

*SETD7* (Supplementary Fig. 4c), consistent with the previously reported CD1 group[22].

We then identified genes differentially expressed across the B cell lineage, from primitive B cells to mature B cells and ultimately to normal and malignant plasma cells. We found four groups of overexpressed genes that defined each stage (Fig. 3b and Supplementary Data 4). The Primitive B group included *SOX4* and *DNTT*, along with several less-investigated genes in terms of lineage, such as *HMGB1* and *HMGB2*, both of which are involved in DNA double-strand breakage[36,37] and might be associated with VDJ recombination. The Mature B group was defined by *CD20* (*MS4A1*) and MHC-associated genes. The third group showed increased expression along the B cell lineage, with high expression in both normal and malignant plasma cells. As expected, ER stress response gene *XBP1* was overexpressed since

plasma cells produce high levels of secreted proteins[38]. The final group showed high gene expression for malignant plasma cells only. Typical genes for this category include *FRZB*, *CD40*, *BIRC3*, and *ZBTB38*. Our discovery of B cell lineage genes is confirmed by the observation of increased expression of MHCII-related genes from primitive B cell to mature B cell stage prior to differentiation to plasma cells. Further, this observation is validated in our independent CyTOF experiment, where the CD38-low, CD45-high, mature population exhibits higher levels of HLA-DQA1 (Supplementary Fig. 5).

We also analyzed single-cell copy number in B and plasma cells and found that 17 out of 21 samples showed chromosome 13 deletion (Fig. 3c and Supplementary Fig. 2d). Complete loss of chromosome 13 is associated with more aggressive malignancy than partial loss, in part because tumor suppressors such as *RB1*

reside there. We identified clusters with deeper chromosome 13 deletion in 83942 Primary, 57075 Relapse-1, and 27522 Relapse-2, indicating possible homozygous deletion in their plasma cells. Clusters with deeper deletion tended to be patient and subpopulation specific, while cells mapping to the same location as normal plasma cells tended to come from multiple patients and showed greater variability, including some with neutral CNV. This suggests deep chromosome 13 deletion is an important feature in determining the overall expression profile of malignant plasma cells.

**Distinct plasma cell subpopulations remain stable during transition from SMM to primary.** We investigated how clonal structure evolves from SMM to primary diagnosis in three patients, 37692, 47491, and 58408 (Fig. 1a). Without exception, we found that plasma cells grouped into two geometrically distinct t-SNE subclusters (subpopulations) in both disease stages (Fig. 4a and Supplementary Fig. 6a, b).

To investigate whether primary plasma cell subpopulations descended from subpopulations present at SMM, we integrated data from the two disease stages and examined how the respective cells cluster. In Patient 58408, we found a good mixture for clusters 1 and 2 from the two stages, which occurred 4.0 years apart (Fig. 4b). We then compared genetic alterations and the expression profiles of these clusters (Fig. 4c), finding clear chromosome 13 loss in cluster 1 of both the SMM and primary stage, while cluster 2 of both stages exhibited normal copy number. Gains on chromosomes 5 and 15 show a similar concordance (Fig. 4c and Supplementary Fig. 6c). This evidence collectively suggests that Primary subpopulation 1 probably descended from SMM subpopulation 1, and likewise for subpopulation 2 at the two time points.

We repeated this analysis in the other two patients (47491 and 37692) (Supplementary Fig. 6d–h) and found the same pattern. In Patient 47491, cluster 2 from SMM matches cluster 1 from primary, and the remaining two clusters are associated with each other. This is illustrated by the slight gain of chromosomes 5 and 15, as well as clusters overlapping in the integrated t-SNE plot (Supplementary Fig. 6d, e). Interestingly, B cell and plasma cell-based trajectory analysis also partially agrees with the expression-based clustering. For example, trajectory state 4 is mainly dominated by cells assigned to cluster 2 from SMM and cluster 1 from Primary, while the other two subpopulations take the other branch and exhibit a higher pseudotime value (Supplementary Fig. 9a–c). For Patient 37692, we also found cluster 1 from SMM and cluster 2 from primary overlapping, while the other two clusters overlapped (Supplementary Fig. 6f).

In Patient 37692, we did not find compelling evidence at the CNV level, possibly due to limited coverage resulting from a low number of plasma cells recovered at the SMM stage. A notable difference regarding Patients 37692 and 58408 is that the dominant subpopulation (the subcluster with more cells) for 58408 at SMM stage remains dominant at primary stage, while the minor subpopulation for 47491 and 37692 at SMM becomes dominant at the primary stage, suggesting differences in the survival/proliferation of distinct plasma subpopulations. Nevertheless, plasma cell population structures are maintained from SMM to primary diagnosis, suggesting a stable population evolution pattern during this transition.

To further understand subpopulation expression profiles, we investigated expression patterns for Patient 58408. We found slightly higher expression of canonical B cell markers CD79A and CD19 in cluster 1 for both time points (Fig. 4c), while expression of plasma cell markers is similar (Fig. 4c), suggesting plasma cell subpopulation 1 represents a more ancestral "B cell-like"

phenotype. Given the presence of chromosome 13 deletion in this cluster, it is possible that malignant transformation of this clone occurs at the B cell rather than plasma cell stage though this could also arise through a reprogramming process. We also conducted an unbiased differential expression analysis and found high expression of JUN, FOS, FOSB, and JUND in cluster 1 (Fig. 4c). Notably, differential expression for FOS and JUN is also found within clusters for the other two patients (Supplementary Fig 6g, h). Expression levels of heat-shock proteins are consistent between the two clusters, though, suggesting such differential expression is not due to stress response (Supplementary Fig 6a). JUN and FOS encode proteins JUN and FOS which dimerize to assemble the AP-1 transcription factor. AP-1 has been implicated in a variety of biological processes, including cell proliferation, differentiation, and apoptosis[39].

We found chromosome 13 deletion in cluster 1 in Patient 58408, suggesting a more malignant phenotype. However, we also detected high levels of JUN and FOS in normal plasma cells, similar to what we found in this cluster. Based on these observations, it is difficult to determine whether high AP-1 activity could be an indicator of malignancy, especially given that the oncogenic role of the AP-1 pathway is very context-dependent[39].

**Dynamic gain and loss of plasma cell subpopulations observed from primary to relapse.** We followed plasma cell populations from the primary diagnosis to relapse and noticed the emergence of distinct plasma cell subpopulations. In each of six patients with primary and relapse time points (27522, 56203, 57075, 59114, 60359, and 81012), we observed two or more t-SNE subclusters of plasma cells, which arose in the context of treatment-related selective pressure (Fig. 1a). Plasma cell subclusters tended to be more similar to (i.e. clustered more closely to) each other than other cell types. The proportion of plasma cells present at the primary and relapse stages varied across patients, with some tumors exhibiting a higher proportion at the primary stage and vice versa; this could reflect sampling variability, patient-to-patient differences in disease progression and treatment efficacies, and/or the snapshot nature of data collection (Figs. 1a and 2a). Next, using single-cell gene expression and copy number changes, we determined the relationship between plasma cell subpopulations at primary and relapse stages. Within a particular patient, subclusters with similar expression and copy number patterns at different time points likely represent the same subpopulation of cells observed over the course of tumor progression. Three patients (81012, 56203, and 27522) illustrate this dynamic population shift in detail.

Patient 81012 displayed variable plasma cell subpopulation dynamics over the course of progressive disease (Fig. 4d–f). At the primary stage, we observed two plasma cell subpopulations (named P.1 and P.2). Later, at relapse, we observed four plasma cell subpopulations (R.1–R.4). In this case, two new plasma cell subpopulations emerged at relapse which had not been observed at the primary stage. Integrated t-SNE mapping showed that the overall expression profiles of P.1 and R.1 match, that P.2 and R.2 match, and that R.3 and R.4 are distinct new clusters (Fig. 4e). Trajectory-based analysis also suggests R.3 and R.4 are mainly present at the end point of state 4 and state 5, respectively (Supplementary Fig. 9d–f). Looking more closely at expression markers, P.1 and R.1 showed elevated expression levels of B cell marker CD79A. P.1, R.1, and R.3 had similar levels of plasma cell markers (SDC1, TNFRSF17, and SLAMF7). For FOS, one component within the AP-1 complex, we found the lowest expression in P.2 and R.2; P.1, R.1, and R.3 exhibit higher expression, while R.4 shows highest expression. We then took a

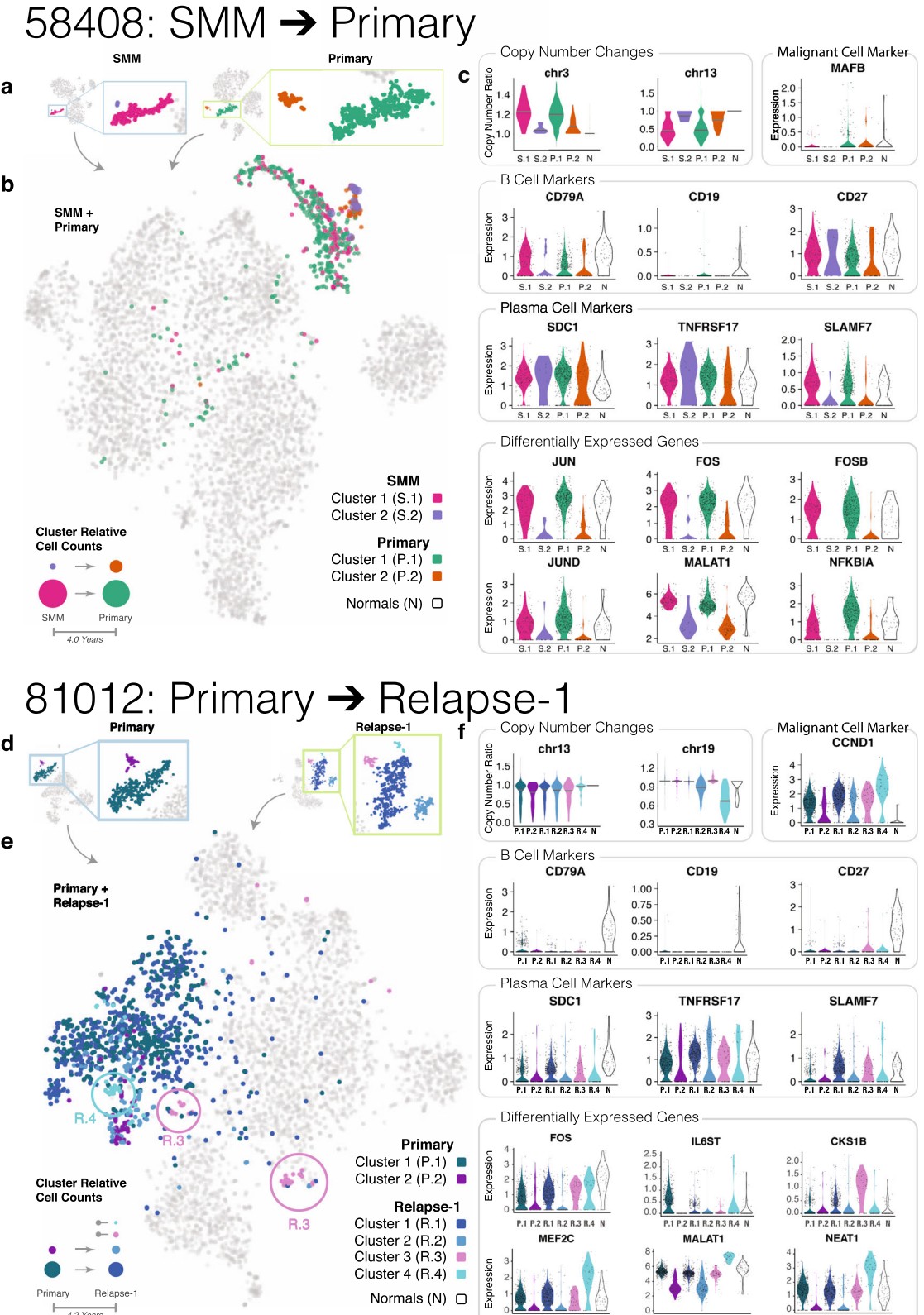

**Fig. 4 Patterns of plasma cell subpopulation shift from SMM to Primary (58408) and from Primary to Relapse (81012). a** Plasma cell t-SNE subclusters for Patient 58408 at SMM and Primary time points. **b** Plasma cell subclusters identified in **a** mapped to the integrated t-SNE of all cells from Patient 58048 SMM and Primary time points. Bottom left: possible explanation for plasma cell subpopulation shift from SMM to Primary. **c** Copy number and expression patterns for plasma cells from different time point subclusters and plasma cells from healthy donors. The first row shows copy number changes and expression of genes associated with genetic alterations detected in Patient 58408. The second and third rows show the expression of B cell markers and plasma cell markers. The last two rows show differentially expressed genes found between the clusters. **d–f** Similar illustrations as **a–c** except for Patient 81012, who progressed from Primary to Relapse-1.

closer look at R.3 and R.4, since the two populations both exhibit high levels of FOS but are newly derived based on mapping in tSNE plot, which suggests the alteration in expression profile apart from FOS upregulation. We found high expression of *MKI67* and *TOP2A*, two proliferative markers for R.3 (Supplementary Fig. 7d). Pathway analysis for the genes specifically upregulated in R.3 points to cell cycle regulation (Supplementary Fig. 7e), which is also consistent with the upregulated proliferative marker expression. This population also exhibits the highest *CKS1B* expression, the overexpression of which promotes myeloma cell growth and survival[40] and is associated with a poorer prognosis[41]. *CKS1B* overexpression could be caused by gain at chromosome 1q21 region, but this was not observed in our analysis, suggesting the change is independent of chromosome alteration. R.4 has the highest expression for *MEF2C*, a transcriptional factor typically regarded as playing a role in muscle cell differentiation[42]. Recently, ATAC-seq profiling suggested *MEF2* family is preferentially enriched in the open chromatin regions in myeloma cells and MEF2C inhibition resulted in reduced myeloma cell growth and survival[43]. At the copy number level, R.4 exhibits chromosome 19 loss, a feature absent in all the other subpopulations (Fig. 4f). Together, the evidence suggests that the newly arisen R.3 and R.4 both exhibit enhanced growth and survival, though through different mechanisms of regulation.

Patient 56203 progressed from the primary stage, with three plasma cell subpopulations (P.1, P.2, and P.3), to the relapse stage, with two plasma cell subpopulations (R.1 and R.2) (Supplementary Fig. 8a). P.1, R.1, and R.2 showed similar levels of chromosome 13 loss, while R.1 and R.2 demonstrated chromosome 17 loss, which distinguished the relapse clusters from the primary clusters. Following drug therapy and ASCT, primary cluster P.1 showed similarity to the two subpopulations present at relapse, while primary clusters P.2 and P.3 appear to have been lost (Supplementary Fig. 8a).

However, tumor subpopulation relationships during disease progression can be more complex than Patients 81012 and 56203 illustrated, as seen in the four time points of Patient 27522 (Fig. 5). The primary time point plasma cells comprise four distinct subpopulations (P.1–P.4) (Fig. 5a). Subpopulations P.1, P.2, and P.3 each show partial loss of chromosome 13, while P4 does not (Fig. 5d). Projection of P.1–P.4 from Patient 27522 onto the integrated cross-sample B cell and plasma cell t-SNE map shows two groupings of P.4, both of which map distantly from P.1 to P.3, largely confirming the original sample-level clustering as well as indicating a high level of population complexity (Figs. 3a and 5b).

We then looked at subpopulations from Remission (RM), Relapse-1 (RL1), and Relapse-2 (RL2) separately from Primary. At Relapse-2, we observed three subpopulations of plasma cells (RL2.1, RL2.2, and RL2.3), with chromosome 13 and chromosome 16 loss in RL2.1, partial loss of chromosome 13 in RL2.3, and t(4;14) translocation in both RL2.1 and RL2.3. RL2.2 remained copy number neutral at chromosome 13 and chromosome 16 (Fig. 5d). Further, we looked for somatic mutations detected from WES data in our scRNA-seq-seq data and noted the occurrence of reference (blue dots) and mutant (red dots) alleles in cells with read coverage. Mutant alleles were detected exclusively in RL2.1, but never in RL2.2 or RL2.3 (Fig. 5e). Somatic events observed in these cells included *NRAS* G13R mutation and t(4;14) translocation (inferred from *FGFR3* and *WHSC1* upregulation) (Fig. 5d and Supplementary Fig. 8b). All three clusters expressed high levels of standard plasma cell markers, such as *SDC1*, *SLAMF7* (*CS1*), and *TNFRSF17* (*BCMA*), while *FGFR3* and *WHSC1* were primarily expressed in the malignant (RL2.1) and the "transitional" malignant (RL2.3) subpopulations. *CD27*, a marker associated with normal plasma cells[44], *CD79A*, a member of the B cell antigen receptor complex, and *CD19*, a marker for B cell development were exclusively detected in RL2.2, supporting the normal "B cell-like" classification (Fig. 5d and Supplementary Fig. 7c). RL2.2 is composed of cells with either high expression of IgA or IgG, while the patient exhibited IgA in isotype identification, which suggests some plasma cells from this subpopulation are normal. These data represent the confirmed observation that combining mutation and CNV/SV mapping and single-cell expression data enables precise identification of cell subpopulations in MM that are either rare or undergoing transitional states, with important clinical implications.

In summary, Relapse-2 comprises three distinct subpopulations, one malignant (RL2.1, with somatic mutations and deep chromosome 13 deletion), one "B cell-like" (RL2.2, with strong B cell marker expression), and one "transitional" (RL2.3, without somatic mutations detected but with shallow chromosome 13 deletion). We then traced the origin of these three subpopulations by integrating Relapse-2 with the Remission and Relapse-1 time points.

Based on an integration of Remission (RM), Relapse-1 (RL1), and Relapse-2 (RL2), we found four groups of cells, which are colored by their time point-specific clusters (Group 1: mostly RL2.1; 2: mostly RL2.2; 3: mostly RL2.3; 4: exclusively RL1.1) (Fig. 5c). Some cells from both Remission and Relapse-1 mapped with RL2.2 (Group 2); it is likely that part of these cells are non-malignant plasma cells based on the expression of IgA and IgG. Likewise, other groups of cells from Remission and Relapse-1 mapped with RL2.3 (Group 3). There was one major subpopulation of cells from RL1 that mapped on its own without any clear connection to the previous or later time points (Group 4). Finally, cells present at Remission mapped with the malignant subpopulation RL2.1 (Group 1). This subpopulation was not seen at Relapse-1, potentially due to low cell count or sampling variability. According to B cell marker expression (*CD79A*, *CD19*, *CD27*), the cell population at Remission shows a "B cell-like" pattern, but the co-clustering of Remission cells to multiple relapse populations indicates there is still some malignancy lurking at Remission. Taken together, one interpretation is there were cells present at remission that evaded treatment and survived to seed the relapse. Expression and copy number changes seen in Relapse-1 split according to their grouping with Relapse-2 and Remission on the integrated map, justifying the use of three clusters for downstream analysis although sample-level clustering did not resolve such clusters (Fig. 5f).

## Haplotype-based mutation analysis increases resolution of clonal evolution subclustering. We examined how cell type and tumor clonal composition change over time and focus here on Patients 58408 and 27522 to illustrate such evolution. In Patient 58408, the population share of CD4+ and CD8+ T cells dropped from being the two most observed cell types at SMM, with monocytes later emerging as the most prevalent cell type at the primary stage (Fig. 2a). Within the plasma cells, we previously described a relatively stable transition of two subpopulations from SMM to Primary (Fig. 4a–c), with both hyperdiploidy (HRD) and chromosome 13 deletion detected at the SMM and primary disease stages. Using a mutation VAF-based approach, we observed little genomic change over the 4.0 years separating SMM and Primary (Fig. 6a, b and Supplementary Data 5). We detected mutated driver genes (*HIST1H1E*-S172T and *NOTCH1*-D2201V) in the main subclone at both time points.

In Patient 27522, we observed NK cells only at the relapse stages (Fig. 2a), and the population share of CD4+ T cells

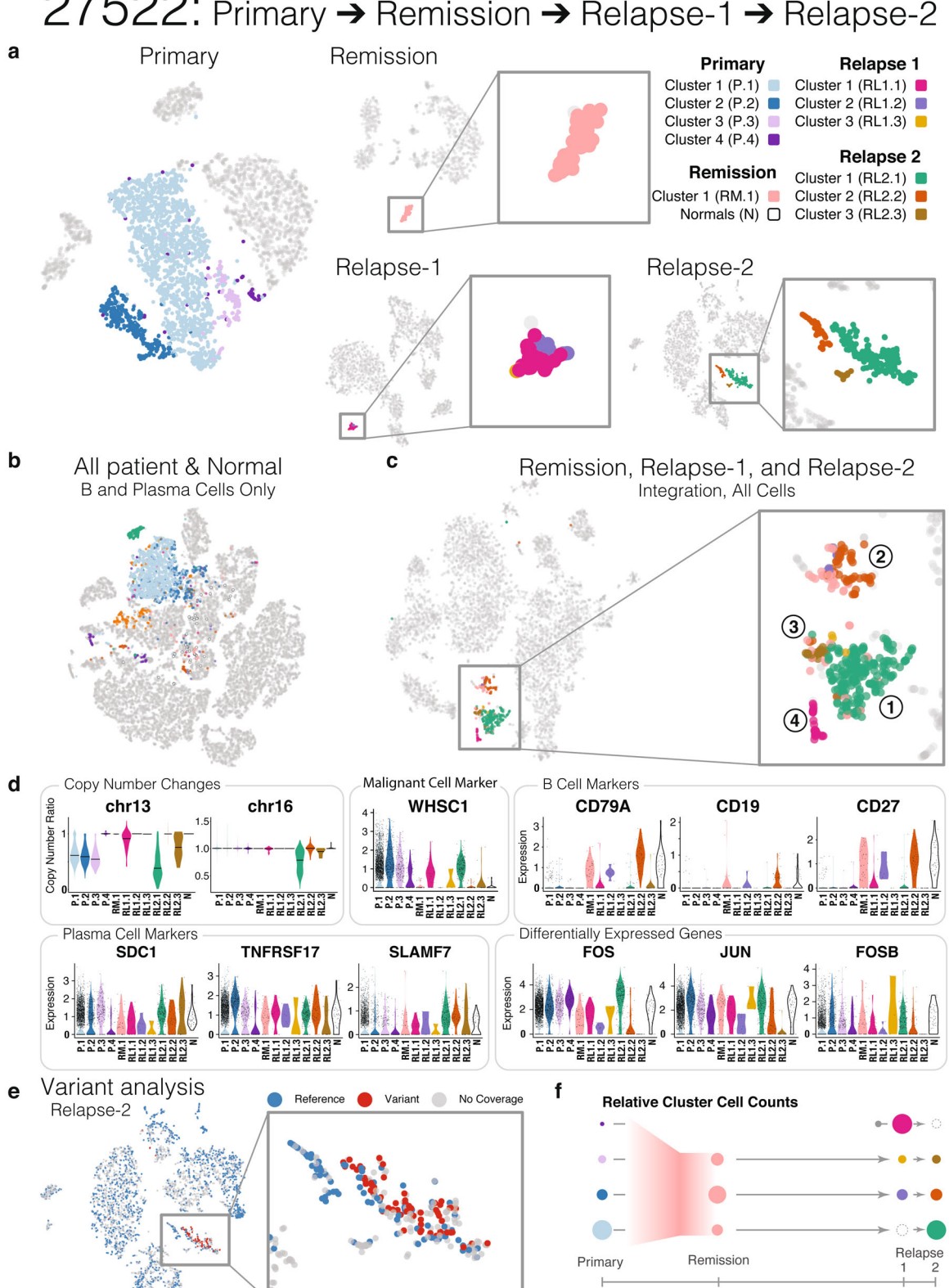

**Fig. 5 Detailed analysis of plasma cell subpopulation shift for Patient 27522. a** t-SNE mapping of plasma cell subclusters for Patient 27522 at Primary, Remission, Relapse-1, and Relapse-2 disease stages. Colors indicate different subclusters within each time point. **b** Plasma cell subclusters identified in **a** mapped to the integrated t-SNE of **b** and plasma cells from all samples plus healthy donors (as in Fig. 3a). **c** Plasma cell subclusters identified in **a** mapped to the integrated t-SNE of all cells from Patient 27522 Remission, Relapse-1, and Relapse-2 disease stages. **d** Subcluster level copy number changes and expression of malignant cell markers, B cell markers, plasma cell markers, and differentially expressed genes. **e** Somatic mutations mapped onto Relapse-2 t-SNE (blue, reference allele only; red, variant allele detected; grey, no coverage). **f** Possible explanation for plasma cell subpopulation shift from Primary to Relapse-2.

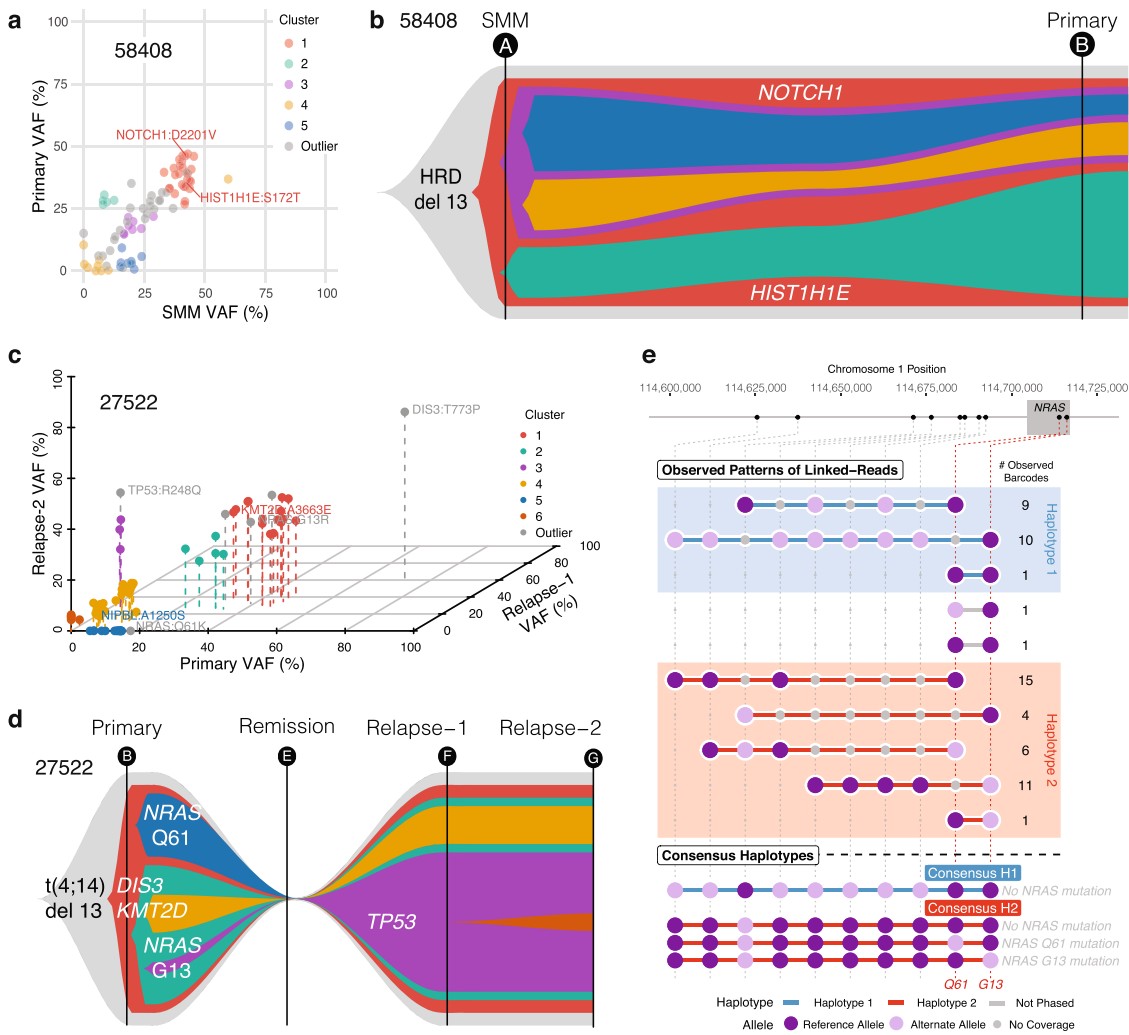

**Fig. 6 Linked-read DNA sequencing maps somatic mutations to germline haplotypes and clonal evolution maps. a** Variant allele frequency clustering of subclonal populations from Patient 58408 SMM and Primary samples. **b** Somatic mutation VAF-based clonality models for Patient 58408. **c** Variant allele frequency clustering of subclonal populations from Patient 27522 Primary, Relapse-1, and Relapse-2 samples. **d** Somatic mutation VAF and haplotype-based clonality model for Patient 27522. **e** Barcode analysis of two *NRAS* somatic mutations showing both mutations occurred on Haplotype 2 did not co-occur, suggesting an independent subclonal relationship. Each set of linked reads represents a particular pattern of support for the two somatic *NRAS* mutations. The number of observed barcodes refers to total barcodes demonstrating the same pattern of *NRAS* somatic mutations.

expanded following remission. The overall proportion of plasma cells observed declined over time with treatment from being a prominent cell type at the Primary stage to later being only a minor cell type. Patient 27522 had one primary and two relapse samples with t(4;14) and del(13q). *NRAS*- G13, *NRAS*-Q61, and *DIS3*-T773 were secondary mutations in the primary sample and *TP53*-R248 was detected at relapse (Fig. 6c, d and Supplementary Data 5). The *TP53* subclone showed higher VAF at relapse compared to other subclones, implying relevance to subclonal expansion. As previously shown with plasma cell subpopulation analysis, we detected somatic mutations only in Relapse-2 cluster 1 (RL2.1, green) (Fig. 5e).

The primary sample of Patient 27522 displayed two *NRAS* hotspot mutations at G13 (chr1:114716124, C>G) and Q61 (chr1:114713909, A>T). We noted that the Q61 mutation was nearly lost (VAF) at relapse and wanted to know if the Q61 mutation occurred in a secondary subclone of the G13 subclone or if G13 and Q61 occurred independently. We utilized 10xWGS[45] to address this question. Compared to previous tumor clonality methods which rely mainly on somatic VAFs[46–50],

linked reads have the advantage of placing variants in their haplotype context and providing direct observations of the relationship between proximal somatic mutations at distances not captured by short reads alone. Surrounding germline variation showed that these two mutations occurred on the same haplotype, but they did not co-occur in linked reads covering both positions ($n = 4$), leading us to interpret that they arose independently in distinct subclones, not sequentially in the same subclone (Fig. 6e).

**Targeted protein assay confirms differential AP-1 expression populations in plasma cells**. To better understand how heterogeneity within a single tumor may be reflected in the functional roles of plasma cell subpopulations, we sought to identify common patterns of pathway enrichment across the subpopulations of multiple tumors ("Methods"). We first divided the plasma cell fractions into a total of 115 discrete subpopulations based on differential gene expression. We then performed pathway enrichment analysis on the differentially expressed genes (DEGs)

of each subpopulation. Correlation analysis of enrichment results resolved three groups with highly similar enrichment profiles (Supplementary Fig. 10a). Group 1 subpopulations share enrichment for pathways related to translation regulation, including nonsense-mediated mRNA decay, and protein targeting to membrane. These findings are consistent with previous work showing the relevance of active translation[27] to myeloma pathogenesis. Group 2 is enriched in various metabolic pathways as well as cellular stress response. These pathways may signify differential interaction with the immune microenvironment. Group 3 shares enrichment of cell cycling and proliferation pathways, which may represent highly proliferative subgroups of their respective tumors. Expression of proliferative markers such as *MKI67* and *TOP2A* are upregulated in the group as well (Supplementary Fig. 10b).

In addition to database-driven pathway enrichment, we identified pathways in which DEGs are known key players. Strikingly, out of 13 cases in which enough plasma cells were detected in each sample to perform subpopulation analysis, we observed 7 cases with tumor subpopulations showing differentially expressed members of the heterodimeric AP-1 transcription factor complex, which we call AP-1-high subpopulations. High expression of AP-1 was not solely associated with a specific chromosome alteration event, but the AP-1-high subpopulation was usually enriched for CNV events. There is also a positive correlation between the expression of single cell and bulk RNA-seq expression for *FOS* ($r = 0.43$) and *JUN* ($r = 0.56$) across samples (Supplementary Fig. 11a). We then evaluated the expression of *FOS* and *JUN* across subclusters and across samples, finding that at least one plasma cell subpopulation within each sample exhibits high expression of *FOS* or *JUN* in all cases, regardless of AP-1 expression differences or the number of subclusters within that sample (Fig. 7a). The preservation of the AP-1-high population across samples suggests this population potentially plays a role in the pathogenesis of myeloma.

Given that external stress could potentially affect AP-1 component expression, we also checked the expression of heat-shock proteins. Case 58408 exemplified the consistent expression patterns of plasma cell subclusters and normal plasma cells seen across patients (Supplementary Fig. 7a and "Methods"). Additionally, we checked single-cell-related QC parameters. Although AP-1-high populations exhibit a higher number of expressed genes, number of unique molecule identifiers, and percentage of mitochondria (Supplementary Fig. 7b), when cells are stratified based on the number of expressed genes, the QC-related AP-1 differential expression pattern is unique to plasma cells (Supplementary Fig. 7c and "Methods"), suggesting the QC parameter differences do not evidently contribute to the observed differential expression of AP-1 components.

Given the frequently observed AP-1 differences within plasma cell populations, we further investigated whether and how differences in the AP-1 pathway could lead to biological differences in plasma cell subpopulations. We performed CyTOF experiments with four additional MM patient samples, three of which had good cell viability. We designed two target panels to separate relevant cell types, quantify signaling pathways (e.g. JAK–STAT, NK-kB), and investigate interleukin activity[51,52]. As expected, we found distinct clusters with differential expression of JUN and FOS (Supplementary Fig. 11b). In fact, a closer look at sample 81,198 indicates the two populations with differential AP-1 expression are evident after t-SNE dimension reduction using only cell surface markers (Fig. 7b).

We then combined results from scRNA-seq and CyTOF experiments for a deeper analysis of AP-1 targets (Fig. 7c). We noticed the expression of *H3F3B* and *ZBTB20*, which are downstream targets of FOS based on ENCODE experiment ENCSR000EYZ[53,54], are concordant with AP-1 expression within plasma cell populations. *H3F3B* encodes H3.3, a variant of histone H3. Ectopic overexpression of H3.3 is sufficient to induce senescence-associated heterochromatin foci (SAHF), an important marker for cellular senescence[55]. *ZBTB20* reportedly plays a role in B cell terminal differentiation; its expression in plasma cell lines induces cell survival and blocks cell cycle progression[56]. Apart from this, we also found that *MCL1*, a marker for survival, and *CDKN1A*, a cell cycle inhibitor, are slightly upregulated in the AP-1-high population, suggesting a potential association between AP-1-high expression and cell survival. Enhanced survival, decreased cell proliferation, as well as the presence of SAHF, all suggest a senescent phenotype for the AP-1 upregulated population. We also found higher expression of IL6ST in the AP-1-high population. IL6ST is a signal transducer shared by IL-6 family cytokine members and is implicated in the progression of various cancer types[57,58]. IL-6, one of the ligands for IL6ST, and IL1B were upregulated in Patients 81198 and 31570. Given that both samples have undergone prior treatment, it is possible that different populations of plasma cells respond to treatment differently by producing differential amounts of cytokines, especially those involved in senescent-associated-secretory profile.

It should be noted that, while FOS and JUN are co-dysregulated for a specific cluster in most cases, there are situations where only one of the molecules is dysregulated while the other one is much less obvious. For example, in sample 83942, where no AP-1 differences among clusters are observed, we found all the clusters exhibit low expression of FOS while JUN expression is high. A more interesting case is for sample 81198, where the AP-1-high population exhibits higher upregulation of JUN compared to FOS. In this sample, the AP-1-high population exhibits downregulated CD138 expression and upregulated IL32 expression compared to AP-1-low population. Hypoxia could downregulate CD138 expression in myeloma cells[59] and induce IL-32 in myeloma cells[60], suggesting AP-1 high population has a more obvious hypoxic signature. Meanwhile, JUN has been shown to stabilize *HIF1A* in a transcriptionally independent manner[61]. It is likely that JUN stabilizes *HIF1A*, promoting the expression of a series of downstream targets, including IL-32, a phenomenon not expected for a cluster where only FOS is high. In summary, different components within the AP-1 complex could play different roles in shaping the downstream effector, contributing to diverse phenotypes of plasma cells.

## Discussion

In this study, we applied a combination of conventional and single-cell technologies to systematically study MM in 14 patients with different treatments at multiple stages of disease progression. We performed scRNA sequencing for ~75K single cells, including both malignant and non-malignant cells, to better understand the transcriptome profiles of these tumors and their interactions with the microenvironment. Varying compositions of cell types over the disease course (e.g. fluctuation of numbers of CD4+ T cell numbers in Patient 59114 discussed above) support the view that the tumor microenvironment is fluid and plays an active role in inter-tumor heterogeneity, as well as disease progression. Patients with the same genetic alterations tended to have both plasma cells and immune cells clustered together. For example, in our two patients with t(11;14), we noted distinct T cell clusters in the tumor microenvironment as well as upregulation of lysine methyltransferase genes *KMT2A* and *KMT2C* in CD8+ T cells. After integrating the data from inferred plasma cells and B cells from healthy donors, we were able to catalog a lineage from primitive B cells to mature B cells and ultimately to normal or malignant plasma cells. Many genes related to this lineage were

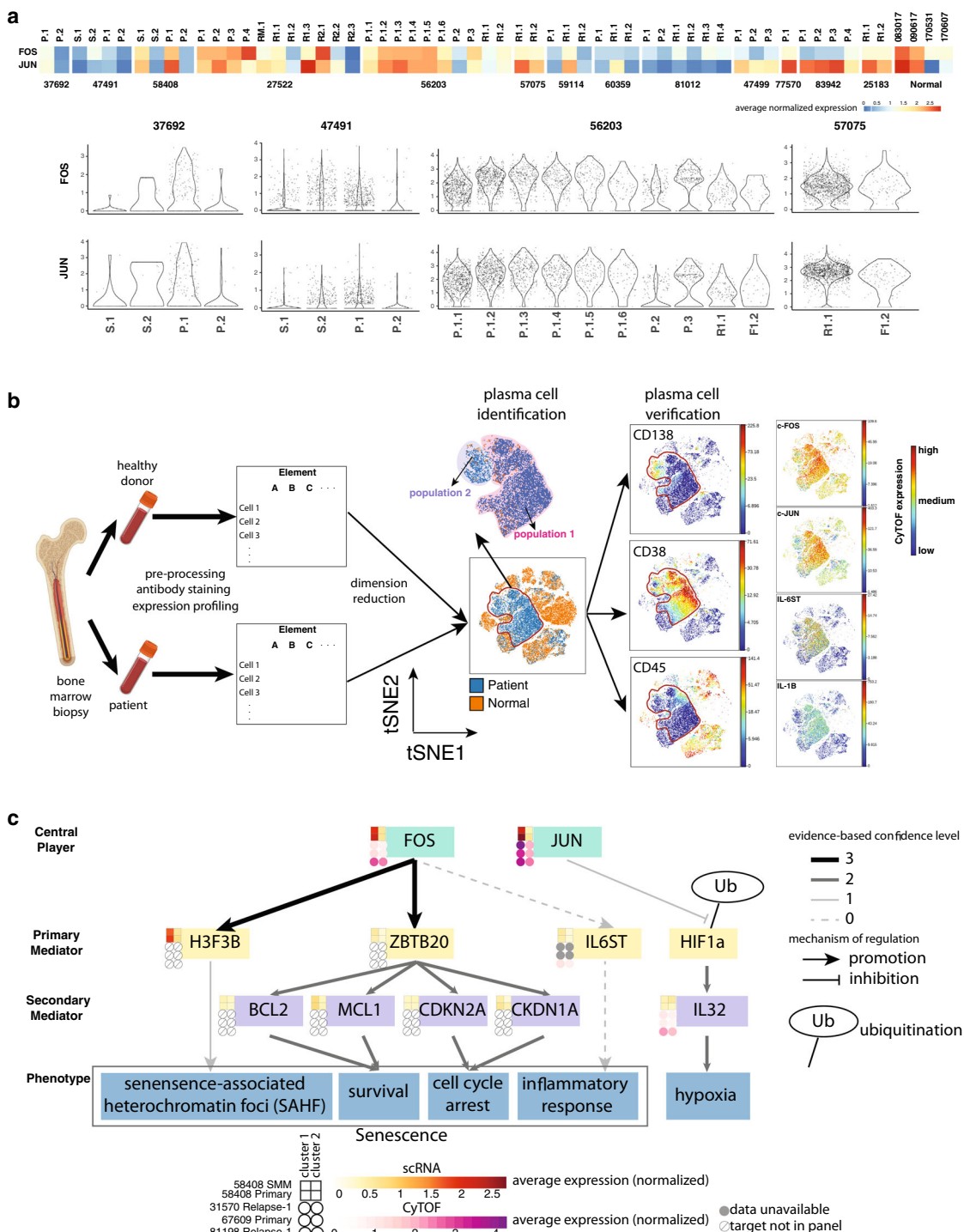

**Fig. 7 AP-1 expression population in plasma cells confirmed by independent cohort. a** AP-1 components expression across plasma cell subpopulations across samples. Upper: average expression for *FOS* and *JUN* for each subpopulation. Lower: violin plot showing the expression patterns of *FOS* and *JUN* for some cases of interest. S SMM, P Primary, RM Remission, R1 Relapse-1, R2 Relapse-2. **b** CyTOF experiment workflow and data analysis. Bone marrow samples from patient and healthy donors are preprocessed, stained for target antibodies of interest, and expression is profiled in parallel. Samples from patients and healthy donors are merged together and visualized with t-SNE. Regions where only patient samples occupy are further checked for CD138, CD38, and CD45 for verification of their plasma cell identity. Expression profile for FOS, JUN, IL-1B and IL-6 within plasma cells are shown. **c**. Proposed mechanism of how AP-1 complex influences the phenotype of myeloma cells. Heatmap beside each gene indicates normalized expression for different populations of plasma cells in Patients 58408 (SMM and Primary), 31570, 67609, and 81198. scRNA-seq expression data, yellow scale; CyTOF expression data, purple scale. Solid arrows indicate the presence of evidence from literature or database. Dashed arrows indicate indirect evidence. Color of solid arrows indicates the confidence level of the evidence of origin. 3, evidence from ChIP-seq database; 2, evidence from myeloma associated literature; 1, evidence from non-myeloma associated literature. Clusters 1 and 2 for each case are manually defined.

identified, including known genes like *XBP1*, as well as additional genes requiring further characterization. The overall result indicates that single-cell transcriptome profiling of B cells and plasma cells could be used to trace the origin of MM, and we identified some patients with plasma cells that exhibit a B cell signature.

We investigated how plasma cell population structure evolves from SMM to primary diagnosis to relapse by integrating somatic alterations mapping, cell lineage marker gene expression, and differential gene expression. Although previous studies have characterized the stability of the SMM to primary transition, we traced specific plasma cell subpopulations across disease stages to illustrate this process and extended the analysis to highlight dynamic changes from diagnosis to relapse. Our analysis delineates the plasma cell subpopulation structure during MM disease progression. By integrating scRNA-seq and genomic alterations, we built plasma cell evolution models representing transitions between disease stages and highlighted co-evolution with the tumor microenvironment. In contrast to malignant cells, non-malignant cells clustered by cell type, independent of their tumor of origin and disease stage. However, detailed characterization of individual immune cell types showed some patients with distinct expression profiles, suggesting a potential interplay between the genomic landscape and an altered microenvironment.

We identified distinct subpopulations of plasma cells in most samples and observed three major patterns of subpopulation shift during disease progression: stable, gain, and loss. Stable pattern is seen in all three patients from SMM to primary, while gain and loss of subpopulations are found from primary to relapse. We extend conventional mutation VAF-based tumor evolution inference models by directly observing subclonal relationships using single-cell and single-molecule mutation mapping. In the future, mutation mapping should provide more useful information as scRNA-seq technology keeps evolving. We believe mutation and CNV mapping carried out in conjunction with gene expression clustering strategies may be generalizable to other cancer types to trace the origins of malignant cells.

Plasma cells from different populations within the same sample usually exhibit differential expression for components within the AP-1 complex, e.g., JUN and FOS. Tracing the co-differentially expressed genes, together with ChIP-seq data analysis, revealed potential downstream targets which contribute to enhanced survival but decreased proliferation of the AP-1-high population. CyTOF experimentation revealed a similar pattern in FOS and JUN expression. The presence of additional DEGs from the CyTOF panel, such as IL6 and IL1B, potentially suggests a greater inflammatory response happening in the AP-1 high population.

Future study designs will enable us to compare greater numbers of patients within the same treatment regimen to better understand effects of treatments on tumor and immune cells. In addition to single-cell transcriptomics, integrating single-cell proteomics will bolster our ability to comprehensively investigate disease progression and treatment response in MM.

## Methods

**Patient cohort**. Fourteen patients with MM, 10 male and 4 female Caucasians, were included in the analysis. All patients were diagnosed and treated at Washington University and provided consent in the written form for the usage of their samples for research purposes. The median age at diagnosis was 63 (range 46–69). Eight patients had IgG isotype, 4 being kappa light chain and 4 being lambda light chain, 2 had IgA kappa isotype, 2 had light chain only disease (1 kappa and 1 lambda), and 2 were non-secretory. Five were International Staging System Stage 1, two were Stage 2, three were stage 3, and four were unreported. The median plasma cell burden by flow cytometry in bone marrow at diagnosis was 24% (range 4–63). By standard fluorescence in situ hybridization (FISH), one patient had t(4;14), three had t(11;14), and two showed del(17p). Four additional patients were included for validation. Two patients have IgG isotype, one being kappa light chain and one being lambda light chain. One has IgA lambda isotype.

One patient has light chain disease (lambda). A more detailed description for patient clinical information could be referred from Supplementary Data 1.

**Processing**. Research bone marrow aspirate samples were collected at the time of the diagnostic procedure. The protocol has been approved by the Washington University Institutional Review Board. All relevant ethical regulations, including obtaining informed consent from all participants, were followed. Bone marrow mononuclear cells (BMMCs) were isolated using Ficoll-Paque. BMMCs were cryopreserved in a 1:10 mixture of dimethyl sulfoxide and fetal bovine serum. Upon thawing, whole BMMCs were used for scRNA-seq (unless otherwise specified), 10xWGS, and RNA-seq, as described below. Plasma cells were separated from a sub-aliquot by positive selection using CD138-coated magnetic beads in an autoMACs system (Miltenyi Biotec, CA) and used for WGS, IDT exome, and RNA-seq, as described below. Skin punch biopsies were performed at the time of the diagnostic bone marrow collection to serve as normal controls for WGS. Although many studies use peripheral blood mononuclear cells (PBMCs) as a control, abnormal B cells and circulating tumor cells frequently contaminate the peripheral blood of patients with MM. Therefore, using PBMCs may lead to omission of genetic events potentially important in disease pathogenesis.

**Single-cell library prep and sequencing**. Utilizing the 10x Genomics Chromium Single Cell 3′ v2 or 5′ Library Kit and Chromium instrument, approximately 17,500 cells were partitioned into nanoliter droplets to achieve single-cell resolution for a maximum of 10,000 individual cells per sample. The resulting cDNA was tagged with a common 16nt cell barcode and 10nt Unique Molecular Identifier during the RT reaction. Full-length cDNA from poly-A mRNA transcripts was enzymatically fragmented and size selected to optimize the cDNA amplicon size (approximately 400 bp) for library construction (10x Genomics). The concentration of the 10x single-cell library was accurately determined through qPCR (Kapa Biosystems) to produce cluster counts appropriate for the HiSeq4000 or NovaSeq6000 platform (Illumina). In all, $26 \times 98$ bp (3′ v2 libraries) or $2 \times 150$ bp (5′ libraries) sequence data were generated targeting between 25 and 50K read pairs/cell, which provided digital gene expression profiles for each individual cell. For all the samples included in this study, only Patient 27522 Relapse-2 was processed with the 5′ Library Kit.

**10xWGS**. The normal skin samples were processed with a standard Qiagen DNA isolation kit resulting in 10–50 kb DNA fragments. In all, 250K tumor cells were processed with the MagAttract HMW DNA extraction kit (Qiagen) resulting in 100–150 kb DNA fragments. Six hundred to 800 ng of normal DNA was size selected on the Blue Pippin utilizing the 0.75% Agarose Dye-Free Cassette to attempt to remove low molecular weight DNA fragments. The size selection parameters were set to capture 30,000–80,000 bps DNA fragments (Sage Science). The resulting size selected DNA from the normal samples and the HMW DNA from the tumor cells were diluted to 1 ng/µL prior to the v2 Chromium Genome Library prep (10x Genomics). Approximately 10–15 DNA molecules were encapsulated into nanoliter droplets. DNA molecules within each droplet were tagged with a 16nt 10x barcode and 6nt unique molecular identifier during an isothermal incubation. The resulting barcoded fragments were converted into a sequence ready Illumina library with an average insert size of 500 bp. The concentration of each 10xWGS library was accurately determined through qPCR (Kapa Biosystems) to produce cluster counts appropriate for the HiSeqX/Nova-Seq6000 platform (Illumina). In all, $2 \times 150$ sequence data were generated targeting 30× (normal) and 60× (tumor) coverage providing linked reads across the length of individual DNA molecules.

**Standard WGS**. Manual libraries were constructed with 50–2000 ng of genomic DNA utilizing the Lotus Library Prep Kit (IDT Technologies) targeting 350 bp inserts. Strand-specific molecular indexing is a feature associated with this library method. The molecular indexes are fixed sequences that make up the first 8 bases of read 1 and read 2 insert reads. The concentration of each library was accurately determined through qPCR (Kapa Biosystems). In all, $2 \times 150$ paired end sequence data generated ~100 Gb per normal and ~200 Gb per tumor sample that lead to ~30× (normal) and 60× (tumor) haploid coverage.

**IDT exome**. A 700 ng aliquot of the existing WGS library was used for the exome capture. Five libraries were pooled at an equimolar ratio yielding a ~3.5 µg library pool prior to the hybrid capture. The library pools were hybridized with the xGen Exome Research Panel v1.0 reagent (IDT Technologies) that spans a 39 Mb target region (19,396 genes) of the human genome. The concentration of each captured library pool was accurately determined through qPCR (Kapa Biosystems) to produce cluster counts appropriate for the NovaSeq6000 platform (Illumina). In all, $2 \times 15$ bp sequence data were generated ~50 Gb per library targeting a mean depth of coverage of 500×.

**RNA-seq**. Total RNA was isolated from ~700K cells utilizing the AllPrep DNA extraction kit (Qiagen). ERCC RNA Spike-In Mix 1 was added to 100–250 ng of total RNA as outlined by the manufacturer (Ambion, Life Technologies). The ERCC control mix is a set of external RNA controls that enable performance

assessment for gene expression experiments. The cDNA library was prepared with the TruSeq Stranded Total RNA Sample Prep with Ribo-Zero Gold kit (Illumina). The concentration of each cDNA library was determined through qPCR (Kapa Biosystems). In all, 2 × 150 reads were generated on the HiSeq4000/NovaSeq6000 instrument (Illumina) generating ~83 million read pairs/sample.

**Dataset description.** The data corpus is comprised of 14 patients having various combinations of sample types, time points, data types, and treatment modalities (Fig. 1a). Most patients have 10xWGS data for skin normal and pre-treatment state, with several having relapse data, as well. Patients 59114 and 81012 underwent relatively long treatment periods before relapse (Supplementary Data 1). Treatment ranges from none for seven patients (three of which have an SMM sample) to multi-cycle regimens of several two-drug and three-drug cocktails, for example in Patient 27522. There are nine patients having at least one time point with both WES and WGS data. Some patients, such as 27522 also have regular whole-exome and whole-genome shotgun data at several time points. All WES and WGS data are generated with CD138+ sorted population (tumor cells) within bone marrows. Two patients have data from a first and a second relapse (Relapse-1 and Relapse-2), with Patient 59114 having an additional complement of pre-/post-transplant samplings. To ensure samples matched across time points, we compared germline VAFs at 24 loci (Supplementary Fig. 1b and Supplementary Data 1).

**Somatic mutation detection.** Somatic variants were called by our Somatic-Wrapper pipeline, which includes four established bioinformatic tools, namely Strelka, Mutect, VarScan2 (2.3.83), and Pindel (0.2.54)[62–65]. We retained SNVs and INDELs using the following strategy: keep SNVs called by any two callers among Mutect, VarScan, and Strelka and INDELs called by any two callers among VarScan, Strelka, and Pindel. For these merged SNVs and INDELs, we applied coverage cutoffs of 14X and 8X for tumor and normal, respectively. We also filtered SNVs and INDELs with a high-pass VAF of 0.05 in tumor and a low-pass VAF of 0.02 in normal. The SomaticWrapper pipeline is freely available from GitHub at https://github.com/ding-lab/somaticwrapper.

**Copy number and structural variation detection.** We used BIC-seq2 (ref. [66]), a read-depth-based CNV calling algorithm to detect somatic CNVs using standard WGS tumor samples and paired skin 10xWGS data (human genome GRCh38 reference). The procedure involves (1) retrieving all uniquely mapped reads from the tumor and paired skin BAM files, (2) removing biases by normalization (NBICseq-norm_v0.2.4), (3) detecting CNV based on normalized data (NBICseq-seg_v0.7.2) with BIC-seq2 parameters set as–lambda=90–detail–noscale–control. In WES data, we used CNVkit (v0.9.4)[67] to compare our tumor samples to a background panel of normals. For scRNA-seq data, we used inferCNV (v0.8.2)[15].

Since we analyzed copy number alteration data from multiple different platforms and varying tumor purity levels, we used five ordered categories to describe copy number changes: deletion < loss < neutral < gain < amplification. The CNV category cutoffs (log2 copy number ratio) were −1, −0.25, 0.2, and 0.7, based on BIC-seq2 and CNVkit documentation. For scRNA based copy number, we transformed the inferCNV results to the log2 scale and set cutoffs at −1, −0.4, 0.3, and 0.7.

Somatic structural variants (SVs) were detected by Manta[68] using tumor/normal sample pairs of standard WGS and paired skin 10xWGS. To filter false-positive SVs, we removed events with somatic score <30 and junction somatic score <30. We used bulk RNA and single-cell RNA data to confirm if translocation events showed overexpression compared with non-translocation samples. We collected translocation and gene expression results relevant to MM based on literature (Supplementary Data 7).

**Analysis of 10xWGS data.** The proprietary Long Ranger system (v2.2.2) from 10x Genomics was used for preliminary analysis, including demultiplexing cDNA libraries into FASTQ files and aligning reads to the human genome reference GRCh38 (GRCh38-2.1.0). To call variants using Long Ranger, we used–vcmode with GATK (version 3.7.0-gcfedb67)[69]. Long Ranger phasing quality metrics were extracted from the summary output file associated with each sample. For haplotype analysis of somatic variants, we relied on phase information of germline variation from surrounding loci on the same set of linked reads.

**Ancestry analysis.** We used a reference panel of genotypes and clustering based on principal components to identify the likely ancestry of our 14 MM individuals, with an additional 856 Multiple Myeloma Research Foundation (MMRF) cases (including 31 multiple time point cases). We randomly selected 10,000 coding SNPs from minor allele frequency >0.02 from the 1000 Genomes Project[70]. From that set of loci, we measured the depth and allele counts of each sample's bam using the tool bam-readcount (version 0.8.0). Genotypes were called using these criteria: 0/0 if reference count ≥8 and alternate count <4; 0/1 if reference count ≥4 and alternate count ≥4; 1/1 if reference count <4 and alternate count ≥8; and ./. (missing) otherwise. After filtering markers with vacancies >5% in our MM samples, 6349 markers were left for analysis. We performed principal component analysis (PCA) on the 1000 Genomes samples to identify the top 20 principal components. We then projected our MM samples onto the 20-dimensional space

representing the 1000 Genomes data. To predict the likely ancestry of our MM samples, we built a random forest classifier using these 20 principal components, which has known ancestry information for each sample. Using an 80/20% split between training and test data, our classifier had 99.6% test accuracy (https://github.com/ding-lab/ancestry). We then predicted the likely ancestry of our MM samples based on this classifier.

**Analysis of bulk RNA-seq data.** Gene expression was estimated using Kallisto (v0.43.1)[71] and gene fusions were detected using STAR-Fusion (v1.4.0)[72]. We used GRCh38_v27_CTAT_lib_Feb092018 from the STAR-fusion website as the human reference and corresponding GENCODE annotation sets.

**Analysis of scRNA-seq data.** For single-cell RNA-seq analysis, the proprietary software tool Cell Ranger (v2.1.1) from 10x Genomics was used for demultiplexing sequence data into FASTQ files, aligning reads to the human genome (GRCh38), and generating gene-by-cell UMI count matrix. The R package Seurat (v2.0) was used for all subsequent analysis[73]. First, a series of quality filters was applied to the data to remove those barcodes which fell into any one of these categories: too few genes expressed (possible debris), too many UMIs associated (possible more than one cell), and too high mitochondrial gene expression (possible dead cell). The cutoffs for these filters were as recommended by the Seurat package. Next, the data were normalized and scaled and dimensional reduction was performed using PCA. The cells were then clustered using graph-based clustering (default of Seurat) approach. Cell types were assigned to each cluster by manually reviewing the expression of marker genes. The marker genes used were *CD79A, CD79B, MS4A1* (B cells); *CD8A, CD8B, CD7, CD3E* (CD8+ T cells); *CD4, IL7R, CD7, CD3E* (CD4+ T cells); *NKG7, GNLY* (NK cells); *MZB1, SDC1, IGHG1* (Plasma cells); *FCGR3A* (Macrophages); *CD14, LYZ* (Monocytes); *FCER1A, CLEC10A* (Dendritic cells); and *AHSP1, HBA, HBB* (Erythrocytes). All cells that were labeled as erythrocytes were removed from subsequent analysis.

**scRNA-seq data integration.** Different scRNA gene expression matrices were integrated using the Seurat (v2.0) R package. We controlled for batch effects using the CCA method and the data were integrated using the top 1000 variable genes from each sample and the first 15 CCs. Cell types were assigned based on manual review of marker gene expression (as described above). Cells with inconsistent cell type assignments between the integrated and individual analyses were filtered out. In some cases, the inconsistencies arose from evident clustering issues (for example, when reviewing marker gene expression, two subclusters were obvious within one cluster). Such instances were manually resolved and the cells were rescued. All differential gene expression analyses were carried out using the FindMarkers function of the Seurat package. The default Wilcoxon test was used and hits with adjusted *p* value < 0.05 were deemed significant.

**scRNA-seq correlation analysis.** After integration, for each cell type, we compared the gene expression to other types to identify the significant highly expressed genes (adjusted *p* value < 0.05 and log fold change >0). Then their average expressions in each sample were calculated. Their pairwise correlations were then estimated.

**Clustering of subpopulations of plasma cells based on pathway enrichment.** We used DEGs (fold change >1.5 and FDR < 0.1) to resolve subclusters in plasma cells for each sample. We then performed pathway enrichment analysis using ReactomePA (available at https://github.com/YuLab-SMU/ReactomePA) on the DEGs of each sample subcluster. This was followed by Pearson correlation across the subcluster-associated enrichment *q*-values (log-transformed FDR) for 910 pathways that were significantly enriched (FDR < 0.05) in at least one subcluster. Finally, we hierarchically clustered sample subclusters by their resulting correlation *R* values to identify pathway enrichment clusters. Highlighted pathways were selected for each pathway enrichment cluster based on prevalence within the cluster and average *q* value of its members.

**10Xmapping.** scRNA data provide an unprecedented resource for studying tumor heterogeneity and clonal evolution. Connecting somatic mutations to individual cells can help to better understand these aspects and have the potential to identify tumor cells which cannot be unveiled purely based on expression data or is difficult to be separated by expression alone. Here, we developed a mapping tool (10Xmapping), which can identify reads supporting the reference allele and variant allele covering the variant site in each individual cell by tracing cell and molecular barcode information in the bam file. The tool is freely available at https://github.com/ding-lab/10Xmapping. For mapping, we used high-confidence somatic mutations from WES data; mutations were combined if data from multiple time points existed.

**Single-cell RNA CNV detection and clustering.** To detect large-scale chromosomal CNVs using single-cell RNA-seq data, inferCNV (version 0.8.2)[15] was used to obtain relative expression intensity of plasma cells in comparison to a set of reference "normal" cells, including B cells, T cells, erythrocytes, NK cells, etc.

Cutoff = 0.1 was used for revealing CNV signals. inferCNV took the raw expression matrix generated from Seurat after several filtering steps, as described above. Subsequently, samples were clustered on inferCNV expression data for 30 genes implicated in MM. Cells for each sample underwent a dimensionality reduction using PCA and t-SNE before clustering. Cells were then clustered with the DBSCAN algorithm. Optimal values for epsilon and minimum points were selected via a grid search. Parameters resulting in the highest Silhouette coefficient were ultimately selected.

**Trajectory-based analysis of B cells/plasma cell lineage**. For trajectory analysis, B and plasma cells as a whole were extracted from each case (across time points), respectively. B cell and plasma cells were then imported into Monocle2 (ref. [74]). Parameters for the analysis were consistent with the tutorial (http://cole-trapnell-lab.github.io/monocle-release/docs/#constructing-single-cell-trajectories), except that (1) cell type is set as the variable for differential expression text and (2) to select genes used for ordering, we set 1e−10 as the $q$ value cutoff. We used the function "plot_cell_trajectory" to visualize B cells and plasma cell subcluster projection in the trajectory. To calculate the proportion of different plasma cell subclusters within each state, B cells and plasma cells that do not belong to any subclusters were removed. The rest of the cells were first normalized by the total number of cells within a time point and then plasma cell subcluster proportions were calculated within each state of interest.

**Quality control cross-check for plasma cells subclusters with different AP-1 expression**. To check whether stress response during sample preparation could shape plasma cell subclusters with different AP-1 component expressions, the expression of a number of heat-shock proteins was checked. To check whether other sources of batch effects could affect the structure of plasma cell subclusters, we checked QC parameters including the number of genes expressed (nGene), the number of unique molecular identifiers (nUMI), and the percentage of mitochondrial cells detected (percent.mito). Based on different levels of those parameters across subclusters, we further divided cells based on numbers of genes expressed (≥1000 or <1000) and checked the AP-1 component expression across different cell types.

**CyTOF**. Thawed bone marrow suspensions were stained with two panels of metal-conjugated antibodies as listed in Supplementary Data 6. The concentrations of the antibodies were either based on the suggestions from the manufacturer (Fluidigm) or based on titration experiments. We used two distinct protocols for cell staining. For panel 1, we included a series of signaling molecules specifically, such as the ones from JAK–STAT pathway and NF-kB pathway[75]. Within this panel, we used three conditions by adding either PBS, PVO4, or TNFα to stimulate samples. Final concentrations for PVO4 and TNFα are 125 μM and 20 ng/mL, respectively. For panel 2, we included a series of interleukins and interleukin receptors. The inclusion of the aforementioned targets are based on their dysregulation in MM[76,77]. We included two components within AP-1 complex, JUN and FOS, in panel 2 as well. To stimulate the production of cytokines, we used three conditions by adding either PBS, R848, or TNFα. Final concentrations for R848 and TNFα are 5 μg/mL and 20 ng/mL, respectively. Protein transporter inhibitors were added to each condition 2 h after the beginning of stimulation, and co-incubation lasted for another 2 h. Gating and data analysis were done using WUSTL Cytobank. Live, single cells are selected by gating out cells/debris with outlier cisplatin and DNA intercalator staining. To perform t-SNE analysis, we used the scaled expression of cell surface marker, including CD34, CD123, CD38, CD3, CD4, CD8, CD19, CD138, CD14, CD16, CD11c, and CD56.

AP-1 targets were identified using ChIP-seq data (ENCODE accession number ENCSR000EYZ)[78,79]. We included four additional myeloma patient samples for expression profiling via CyTOF experiment. For each CyTOF run, a sample from a healthy donor would be included. Expressions of cell surface markers are used for t-SNE. Cells from patient samples which do not overlap those from healthy donors on t-SNE plot are further checked for their expression of CD138, CD38, and CD45. Accordingly, the qualified cells are termed as plasma cells.

**Subclonal analysis**. The R package SciClone[48] algorithm was used to define clonal architecture, and tumor phylogeny was illustrated using Fishplot[80].

**Reporting summary**. Further information on research design is available in the Nature Research Reporting Summary linked to this article.

## Data availability

All sequencing data (10xWGS, WGS, WES, Bulk RNA-seq, and scRNA-seq) used in this study can be accessed at the NCBI under accession code PRJNA694128. CyTOF data have been deposited with the FlowRepository under accession number FR-FCM-Z3EP. For ancestry analysis in Supplementary Fig. 1b, data were also provided by The Multiple Myeloma Research Foundation (MMRF) CoMMpass (Relating Clinical Outcomes in MM to Personal Assessment of Genetic Profile) Study (NCT01454297). The MMRF CoMMpass study could be accessed with dbGaP Study Accession: phs000748.

## Code availability

For 10Xmapping pipeline, code can be accessed at https://github.com/ding-lab/10Xmapping[81]. For ancestry prediction, code can be accessed at https://github.com/ding-lab/ancestry[82].

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

## Acknowledgements

We thank the patients, families, and professionals who have contributed to this study and the Multiple Myeloma Research Foundation CoMMpass Study. This study has been supported by the Paula C. and Rodger O. Riney Blood Cancer Research Fund to R.V. and L.D. and NCI U24CA211006 and U2CCA233303 funds to L.D. We also thank Daniel Zhou Cui for pathway analysis and manuscript feedback, and Alla Karpova and Sohini Sengupta for manuscript feedback.

Clipart of bone marrow and sample tubes from Fig. 7b are created with BioRender. com.

## Author contributions

L.D., J.F.D.P., and R.V. led project design. L.D. led data analysis and interpretation. M.A.F., C.C.F., R.S.F., J.K., D.R.K., T.J.L., R.V., and C.J.Y. curated samples and generated sequence data. R.L., Q.G., S.C., S.M.F., E.P.S., H.S., S.S., A.W., and L.Y. developed data processing and analysis pipelines. R.L., Q.G., S.C., S.M.F., J.S.F., J.O.N., M.P.R., E.P.S., H.S., S.S., M.C.W., and L.Y. performed data analysis and quality control. R.G.J. led figure generation. R.L., J.T.W., and J.F.D.P. performed CyTOF experiments. R.L., J.F.M.M., S.C., S.M.F., Q.G., E.P.S., H.S., S.S., M.A.W., and L.Y. generated figures. L.D., R.L., S.M.F., Q.G., S.R.G., R.G.J., H.S., S.S., S.C., and M.C.W. wrote the manuscript. J.F.D.P., T.J.L., S.T.O., and R.V. provided critical review of the manuscript.

## Competing interests

The authors declare no competing interests.
