## [Peer Review File · Nature Communications]

REVIEWER COMMENTS

Reviewer #1 (Remarks to the Author): Expert in multiple myeloma genomics

Liu and colleagues have revised their previous manuscript, which I previously reviewed for [redacted]. Some of the analyses have changed, but unfortunately the paper retains the same fundamental weaknesses.

MAJOR CRITICISMS

- 1) There is no hypothesis and a complete lack of a central story. There is not even any real hypothesis generation.
- 2) The manuscript consists of a series of observations about gene expression or cell clusters, which usually only apply to a subset of the samples. In some cases, these observations are confined to samples with a particular feature (e.g the CCND1-activating translocations). However, the numbers are too small to draw any conclusions about whether the associations are genuine or are a chance finding. In other cases, the authors describe a pattern that they have observed in a few of the samples, but not the others. It is hard to know what to conclude from this, apart from the (well-known) fact that myeloma is a heterogeneous disease.
- 3) There is very little attempt to link the findings to the biology of myeloma. Where biological interpretations are made, they are speculative and no or minimal attempt is made to test the hypotheses.
- 4) Some of the figures and explications are very hard to follow.
- 5) The real novelty that I see in this dataset is the inclusion of microenvironmental single cell sequencing data. However, these are mostly ignored. There may be a really interesting story pertaining to immuno-oncology hidden within, but there is no real attempt to pursue this.
- 6) Ultimately, the authors neither convince me that they have a particularly novel analytical approach to single cell data, nor that they describe any new insights into myeloma biology, let alone cancer in general.
- 7) I accept that the data have value as a resource. However, I could find no entry under the "Data availability" heading. It is essential that the raw data be made public prior to publication.

SPECIFIC EXAMPLES OF THE ABOVE

Practically every result in the manuscript can be criticized in the same way...

Page 6 – VAF changes are seen during disease progression in patients 27522 and 59114. Can we conclude that this is a general phenomenon? Why is this important?

Page 7 – Hubs for a mutational network in 25722 R.2 sample. What are these? Are they seen elsewhere? Why are they important?

Page 8 – Clustering correlation between SMM and primary samples in patients 47491 and 58408 with lower correlation between primary samples and relapse. There are too few samples to draw any conclusions. However, assuming this is a general phenomenon, what is the biological relevance?

Page 8 – Correlation of clustering by translocation. Are there sufficient samples to claim that this is a broad phenomenon? If so, what is the relevance?

Page 8 – Plasma cells cluster by patient, non-PC by cell type. This is hardly surprising. Of the two SMM samples, there appear to be very few PC in 58408. Is this correct?

Page 8 – Differential expression analyses. Conclusions are drawn from only a few cases: two SMM and one remission. Difficult to draw any conclusions here. Relevance? There are six relapse samples, but only three (for IL1R2) or one (for IL1B) are shown in the figure. Why is this? What about the other relapse samples? Does this in fact indicate that the phenomena are not general? Why do the findings matter?

Page 8/9 – Three cases examined for microenvironmental cells. There is no consistent pattern. What is the biological relevance?

Page 9 – Outlier analysis on two patients. The same problem. Can we draw general conclusions about t(11;14)? What is the relevance?

Page 10/11 – B-cell lineage analysis. A purported association between MS4A1 expression in PCs of patients 56203, 83942, 77570 and CCND1 regulation. What about patient 81012, which has a t(11;14) but very low MS4A1? Why doesn't this show the pattern? What is meant by "aberrant CCND1 regulation according to FISH"? It is not a translocation and there is no overexpression, so what are we talking about? The authors really cannot draw any conclusion about relationship between CCND1 and B-cell phenotype from their data. Even if they could, they still fail to explain why this relevant. If they were speculating about pre-germinal centre origins of myeloma or cells that manifest resistance, this might be interesting, but I cannot see specific evidence of this in the data that they show.

Page 11 – Having been shown four levels of B-cells, we are not told why this is important.

Page 11/12 – Deep chromosome 13 loss in three patients. From the manuscript: "Clusters with deeper deletion tended to be patient and subpopulation-specific, while cells mapping to the same location as normal plasma cells tended to come from multiple patients and showed greater variability". This is very vague. It is not formalized or quantified. Why does it matter?

Page 12 – Investigation of clonal structure. Only three samples. Generalizable? Relevant?

Page 14 – The authors describe two t-SNE clusters in each of primary and relapse samples of six patients. Here, at last, we might have numbers that allow us to generalize a finding. However, we should also presumably see a similar two clusters in all the other primary samples if this is a general phenomenon – I don't think this is shown. Assuming that this is the case, what is the situation in SMM? Is there just one cluster in SMM then two in primary myeloma or are there always two clusters? Ultimately, we are still left with the question: why does this matter for myeloma biology?

Page 14/15 – Proportion of PCs varies between primary and relapse (but not always?). Both increases and decreases are seen. What does this tell us?

Page 15 – Patient 81012 subpopulation dynamics. “Trajectory-based analysis also suggests R.3 and R.4 are mainly present at the end point of state 4 and state 5, respectively.” What exactly are states 4 and 5, apart from some arbitrary point in pseudotime? Why is this important? More confusingly, we are told first that P.2/R.2 have the lowest expression of FOS. We are then told that P.1, R.1, and R.2 exhibit higher expression. So does R.2 exhibit higher or lower expression? Similarly, R.4 is stated to have the highest expression, but we are then told that R.3 and R.4 have similar expression of FOS to R.1. R.4 has chr 19 loss, but we are not told why this matters (chromosome 19 loss is not one of the more common cytogenetic findings in myeloma). The conclusion is that R.3 and R.4 have enhanced growth and survival, but I don't see what data tell us that. Are they basing this on overexpression of CKS1B and MEF2C, respectively? If so, this is a bizarre conclusion. Surely there are multiple other factors that can promote growth and survival that may be expressed slightly higher and/or in combination by P.1 and P.2 (but not making the threshold for differential expression). Again, even if true, why does it matter? Can we say that it is a universal/near-universal phenomenon?

Page 16 – Similar analysis for 56203. (1) They claim that P.1 is closest to the relapse samples, but extended data figure 8a is really hard to follow and it is not clear how they concluded this. (2) Assuming, though, that they are correct in this, how many cells exhibited chr 17 loss in P.1. Were they present before treatment? (3) Surely p53 is the best candidate for a tumor suppressor on chr 17. Why are we not shown p53 expression in the various samples? (4) And, again, what is the biological relevance of all this?

Page 16/17 – The analysis of patient 27522 is poorly written and hard to follow. They conclude by stating that there are “important clinical implications” but don't tell us what these are. The statement: “Taken together, one interpretation is there were cells present at remission that evaded treatment and survived to seed the relapse” seems to be a remarkably anodyne conclusion; any clinician who has treated myeloma knows this already.

Page 21 – It is not clear how fig. 7A supports the text, which is not easy to follow. Are the authors drawing their conclusions on the role of AP-1 in myeloma from fig. 7A (which does not support the conclusion) or from samples 58408 and 81012, which is only two patients?

Fig. 7B shows some correlation between FOS/JUN expression and a few of their known targets (IL6ST and IL1A). This is entirely unsurprising, but doesn't really tell us anything.

Fig. 7C This shows a possible correlation between FOS and H3F3B for only four samples and no or minimal correlation between FOS or JUN and other potential targets. The CyTOF data appear to be mostly missing. I am not sure what we can conclude from this. In terms of the bottom of the figure (i.e. the functional consequences), this is speculative. Many of the proteins may be pleiotropic and, as noted, many of the correlations appear weak. Surely some of these proposed functional consequences are potentially testable, e.g. are there any gene signatures or expression markers of SAHF that could be examined (with the caveat that we are only looking at four samples).

Reviewer #2 (Remarks to the Author): Expert in single-cell RNA-seq, biostatistics, and bioinformatics

The authors have addressed my concerns from the last round of reviews and I have no more reservations. As I wrote last time, this study is done on a small cohort that is very heterogeneous, and most of the results obtained are inconclusive. But, I appreciate the reviewers' point that multiple myeloma is an extremely heterogeneous disease, and it is not realistic right now to conduct a study of the scale necessary to draw integrative general conclusions. I think this is a nice demonstrative study, with successful integration across multiple omics data types, that provides a data resource and initial exploration upon future large scale studies can be based.

Reviewer #3 (Remarks to the Author): Expert in single-cell RNA-seq and tumour microenvironment

The authors have addressed all of my comments in great detail. They have thoroughly reanalysed the single-cell RNA-seq data and included many new analyses of the tumour microenvironment as well as trajectory analysis of the plasma cells, which is a tremendous effort. However, even though the study provides a unique longitudinal dataset and represents a valuable resource, the more detailed analyses still haven't resulted into common trends and significant insights about the progression or treatment of myeloma. The samples are very heterogeneous in terms of their mutational and transcriptional landscape. The manuscript might be more suitable to be published as a resource.

Reviewer #1 (Remarks to the Author): Expert in multiple myeloma genomics

Reviewer 1: Liu and colleagues have revised their previous manuscript, which I previously reviewed for [redacted]. Some of the analyses have changed, but unfortunately the paper retains the same fundamental weaknesses.

MAJOR CRITICISMS

Reviewer 1: There is no hypothesis and a complete lack of a central story. There is not even any real hypothesis generation.

Authors: *We appreciate the referee's point and feel that the manuscript was less than clear in framing our hypothesis and main story. We have remedied this issue in the revision. Summarizing, the key hypothesis is that dynamic population shifts are exhibited in both the malignant plasma cells and the surrounding microenvironment during the disease progression process and that these shifts should be discernable at the single-cell level using scRNA technology. Pursuing this hypothesis, we collected a longitudinal set of myeloma patient samples that encompass multiple disease stages and various treatment modalities. By integrating genomics and single cell mapping, we were able to track plasma cell subpopulations across disease stages, finding they fall into three patterns: stability (from SMM to primary diagnosis) and either gain or loss (from primary diagnosis to relapse). We also found that the expression of AP-1 complex is one of the key features that determines the subcluster structure of plasma cells and this is also validated by CyTOF-based protein analysis.*

Reviewer 1: The manuscript consists of a series of observations about gene expression or cell clusters, which usually only apply to a subset of the samples. In some cases, these observations are confined to samples with a particular feature (e.g the CCND1-activating translocations). However, the numbers are too small to draw any conclusions about whether the associations are genuine or are a chance finding. In other cases, the authors describe a pattern that they have observed in a few of the samples, but not the others. It is hard to know what to conclude from this, apart from the (well-known) fact that myeloma is a heterogeneous disease.

Authors: *We understand the referee's criticism, but would point to the inherent and unavoidable difficulties of gathering large, longitudinal MM cohorts for basic research. Indeed, this would be the first published longitudinal genomics study, as far as we can tell from searching the literature.*

Our sample cohort encompasses the relevant heterogeneities across the genomic landscape, disease stages, and treatment regimes (Figure 1). The rationale for our sample collection strategy is to prioritize the longitudinal survey nature of the study above the consistency of having the complete complement of fully-powered, stratified groups over all classes of genomic alterations and their treatment regimes. The latter design is not possible at this time. For instance, even the most highly mutated driver genes like KRAS only appear in ~20%

patients¹, meaning, for example, that obtaining 50 consistent cases (80% power) would require a sample cohort of >250 patients. Each element of heterogeneity implies similar-sized or even larger patient samples over the various strata. Although our cohort is small, the range of mutations suggests our data are representative of the general myeloma population, which tends to undergo diverse treatment regimens depending on patient-specific factors.

With this rationale, the aim for this study has been to 1) serve as a model of how to longitudinally integrate data from the genomic level, RNA expression level, and the microenvironment side in order to more comprehensively investigate the disease progression of multiple myeloma, and 2) pinpoint observations of note from our cohort, which should serve as a springboard for future clinical studies where greater homogeneities of patient profiles and treatments and fully-powered, longitudinal sample cohorts become available.

Reviewer 1: There is very little attempt to link the findings to the biology of myeloma.

Where biological interpretations are made, they are speculative and no or minimal attempt is made to test the hypotheses.

Authors: We appreciate the reviewer's careful reading of the manuscript. As was mentioned above, the limited sample size within this study precludes definitive conclusions. Given the provisional nature of our observations, we do not feel there is not enough evidence to motivate a deep dive into linking findings to myeloma biology. Rather, we anticipate that this undertaking will fall to future, larger-scale studies of myeloma.

As to specific cases, namely 56203 and 81012, for which the reviewer raises concerns, the revision adds new analysis (expanded in the SPECIFIC EXAMPLE part below; refer to page 14-19 in response letter) as another layer for validating our findings. We are not certain whether the observations we made for either of the two cases have a more general relevance to other myeloma cases; however, the subcluster structure within each time point for each case should be more clear with the additional analysis.

Reviewer 1: Some of the figures and explications are very hard to follow.

Authors: The referee touches upon an important point. We have carefully re-edited our text accordingly to clarify our findings. We also expanded and clarified discussion for parts identified by both the reviewer and the editors, especially within the SPECIFIC EXAMPLE part. Overall, we feel the readability of the revised manuscript should be acceptable.

Reviewer 1: The real novelty that I see in this dataset is the inclusion of microenvironmental single cell sequencing data. However, these are mostly ignored. There may be a really interesting story pertaining to immuno-oncology hidden within, but there is no real attempt to pursue this.

¹ Bezieau, S. et al. High incidence of N and K-Ras activating mutations in multiple myeloma and primary plasma cell leukemia at diagnosis. Hum Mutat 18, 212-224 (2001).

Authors: *We appreciate the reviewer's suggestion and have expanded our treatment of this aspect of our work. We performed a cell-to-cell interaction analysis using cellphonedb² (<https://github.com/Teichlab/cellphonedb>), focusing on tumor interactions with cells from myeloid (Reviewer Response Letter Figure 1a) and lymphoid (Response Letter Figure 1b) lineages. We also investigated the specific interactions between myeloid and lymphoid lineages (Response Letter Figure 1c).*

Several interactions are worth noting. For example, between the myeloid population and tumor cells, we detected a significant interaction between TNFSF13B (myeloid) and TNFRSF17 (Plasma), the former coding for B-Cell-Activating Factor (BAFF) and the latter for B cell maturation antigen (BCMA). BAFF improves myeloma cell survival by upregulating anti-apoptotic protein expression³ and macrophage-derived BAFF seems to confer bortezomib resistance via Akt and NF- κ B pathways in myeloma⁴. Conversely, BCMA is a CAR-T target that is being actively pursued in myeloma treatment⁵. It is possible that enhanced survival of myeloma cells conferred by BAFF is mediated by BCMA and that targeting their interaction would be therapeutically beneficial.

We also found significant correlation between CCL4 (CD8+T cells and NK cells) and GPRC5D (Plasma cells), the latter a myeloma-antigen candidate⁶, recapitulating findings in IMEx⁷ and IntAct (www.ebi.ac.uk). Experimental validation for this interaction is based on two hybrid array screenings⁸. CCL4 (a.k.a. MIP1- β) is an important chemokine mediator of physiological homeostasis via recruitment of regulatory T cells⁹. Regarding myeloma, CCL4 secretion helps promote development of osteolytic lesions via interaction with its cognate

² Efremova, M., Vento-Tormo, M., Teichmann, S. A. & Vento-Tormo, R. CellPhoneDB: inferring cell-cell communication from combined expression of multi-subunit ligand-receptor complexes. *Nat Protoc* 15, 1484-1506 (2020).

³ Neri, P. et al. Neutralizing B-cell activating factor antibody improves survival and inhibits osteoclastogenesis in a severe combined immunodeficient human multiple myeloma model. *Clin Cancer Res* 13, 5903-5909 (2007).

⁴ Chen, J. et al. BAFF is involved in macrophage-induced bortezomib resistance in myeloma. *Cell Death Dis* 8, e3161 (2017).

⁵ Yu, B., Jiang, T. & Liu, D. BCMA-targeted immunotherapy for multiple myeloma. *J Hematol Oncol* 13, 125 (2020).

⁶ Smith, E. L. et al. GPRC5D is a target for the immunotherapy of multiple myeloma with rationally designed CAR T cells. *Sci Transl Med* 11, eaau7746 (2019).

⁷ Orchard, S. et al. Protein interaction data curation: the International Molecular Exchange (IMEx) consortium. *Nat Methods* 9, 345-350 (2012).

⁸ Luck, K. et al. A reference map of the human binary protein interactome. *Nature* 580, 402-408 (2020).

⁹ Bystry, R. S., Aluvihare, V., Welch, K. A., Kallikourdis, M. & Betz, A. G. B cells and professional APCs recruit regulatory T cells via CCL4. *Nat Immunol* 2, 1126-1132 (2001).

receptor CCR5¹⁰. This interaction could shed light on the apparently complex way that CCL4 is associated with myeloma biology.

Another noteworthy interaction is between NCR3 from NK cells and BAG6 from a variety of cell types, including monocytes, macrophages, dendritic cells, NK cells, and plasma cells. In chronic lymphocytic leukemia (CLL), tumor cell-released soluble BAG6 inhibits NK cell cytotoxicity by engaging with NCR3 on NK cells¹¹¹². Whether this behavior occurs in MM remains to be established. Other examples include interactions between CLEC2B from plasma cells and KLRF1 from NK cells and LGALS9 from macrophage/monocytes with CD47/CD44 from T/NK cells. KLRF1 is an activating homodimeric C-type lectin-like receptor involved in cytokine release¹³, while CD47 is an immunosuppressive checkpoint¹⁴.

¹⁰ Abe, M. et al. Role for macrophage inflammatory protein (MIP)-1alpha and MIP-1beta in the development of osteolytic lesions in multiple myeloma. *Blood* 100, 2195-2202 (2002).

¹¹ Reiners, K. S. et al. Soluble ligands for NK cell receptors promote evasion of chronic lymphocytic leukemia cells from NK cell anti-tumor activity. *Blood* 121, 3658-3665 (2013).

¹² Pittari, G. et al. Restoring Natural Killer Cell Immunity against Multiple Myeloma in the Era of New Drugs. *Front Immunol* 8, 1444 (2017).

¹³ Kuttruff, S. et al. NKp80 defines and stimulates a reactive subset of CD8 T cells. *Blood* 113, 358-369 (2009).

¹⁴ Tong, B. & Wang, M. CD47 is a novel potent immunotherapy target in human malignancies: current studies and future promises. *Future Oncol* 14, 2179-2188 (2018).

Response Letter Figure 1. Summary of cell-cell interactions between (a) tumor cells and myeloid populations; (b) tumor cells and lymphoid populations; (c) lymphoid and myeloid populations. Interactions of interest are highlighted in red.

Reviewer 1: Ultimately, the authors neither convince me that they have a particularly novel analytical approach to single cell data, nor that they describe any new insights into myeloma biology, let alone cancer in general.

Authors: *We acknowledge the reviewer's concern for the novelty of this study, but would make the following points:*

First, we combined multi-omics data from the genomic and transcriptomic side longitudinally. Specifically regarding scRNA, we integrated multi-layer information based on mutation mapping (developed internally for this study; <https://github.com/ding-lab/10Xmapping>), inferred copy number profile and expression profile, and found cell subclusters that are either preserved or lost as disease progresses. This kind of longitudinal genomics analysis is novel for MM.

Second, a detailed analysis for longitudinal samples from primary diagnosis and relapse time point identifies unique characteristics of some of these populations from the relapse stage, which we discuss in the following pages (refer to page 14-19 in response letter). We anticipate that future work will utilize a very similar strategy as what we have developed here, except with increasingly larger sample sizes in order to establish whether the patterns observed here hold generally and whether there are more possibilities reflecting disease heterogeneity.

Third, we identified plasma cell clusters that are characterized by differential AP-1 expression levels. While previous studies have emphasized the importance of AP-1 expression in myeloma proliferation and drug resistance¹⁵, this is the first study, to the best of our knowledge, that shows FOS/JUN differential expression co-exists within individual samples. This observation could open the door to future work exploring the biological roles of both the AP-1-high and AP-1-low populations in parallel.

Reviewer 1: I accept that the data have value as a resource. However, I could find no entry under the "Data availability" heading. It is essential that the raw data be made public prior to publication.

Authors: *We acknowledge the reviewer's concern regarding making data public. We are currently working on public SRA submission and expecting the data to be public soon.*

SPECIFIC EXAMPLES OF THE ABOVE

Practically every result in the manuscript can be criticized in the same way...

¹⁵ Fan, F. et al. The AP-1 transcription factor JunB is essential for multiple myeloma cell proliferation and drug resistance in the bone marrow microenvironment. *Leukemia* 31, 1570-1581 (2017).

Reviewer 1: Page 6 – VAF changes are seen during disease progression in patients 27522 and 59114. Can we conclude that this is a general phenomenon? Why is this important?

Authors: *Changes within the tumor of variant allele frequencies over time reflect clonal/subclonal evolution. In 27522, for example, TP53 VAF (based on WES) increased from 0.4% at the primary state to 33.1% at relapse-1, then to 42.6% at relapse-2. This increase means that the minor subclone with TP530R284Q has become resistant to treatment and expands over disease progression. VAF for NRAS-Q61K is 17.1% at the primary stage but drops to 0.6% in relapse-1 and then to 0% in relapse-2, indicating that the subclone containing this variant is eliminated by treatment.*

For 59114 however, not all samples were sorted, meaning VAF could reflect either tumor subclonality OR differences in sample tumor fraction. Because of this ambiguity, we have removed the text describing VAF changes for 59114.

Our varied sampling strategy and the inherent patient diversity make it difficult to conclude whether observed VAF changes are a general disease phenomena. However, assessing VAF changes sheds light on how resistant tumor subclones may grow to dominance over the course of treatment. This is a well-established model for cancerous relapse. Examining VAF changes closely may help guide treatment strategies in combating disease persistence.

Reviewer 1: Page 7 – Hubs for a mutational network in 25722 R.2 sample. What are these? Are they seen elsewhere? Why are they important?

Authors: *We appreciate the reviewer catching this detail. “Mutational network” is not an accurate term in this context. We are only referring to mutations co-occurring in the same cells. We have clarified the text accordingly.*

Reviewer 1: Why are they important?

Authors: *Mutation mapping with single-cell data reveals combinations of co-occurring mutations in the same cells, thus giving a better picture of tumor subclonality than VAF alone. It also can determine gene expression differences between subclones that are characterized by different mappings. Currently the number of mapped mutations is small, but technological improvements in snRNA-seq will push future expansions.*

Reviewer 1: Are they seen elsewhere?

Authors: *For 27522 R.2, we were able to map 7 variants (out of 48 detected in bulk analysis) to 63 plasma cells in single-cell data (Supplemental Table S2). For other samples, we observed either fewer mapped variants or fewer plasma cells bearing such variants and were thus unable to find similar patterns of mutation co-occurrence in those instances. This very likely reflects technological limitations.*

Reviewer 1: Page 8 – Clustering correlation between SMM and primary samples in patients 47491 and 58408 with lower correlation between primary samples and relapse. There are too few samples to draw any conclusions. However, assuming this is a general phenomenon, what is the biological relevance?

Authors: *For cases where SMM, primary, and relapse samples are available, we observe a higher degree of similarity between SMM and primary than between primary and relapse. The presence of subsequent timepoints is in itself a suggestion of higher-risk SMM (having indeed progressed to malignancy) and the dissimilarity between primary and relapse samples suggests treatment-induced alterations in the tumor population. A takeaway here is that comparison between tumor expression profiles for primary-diagnosed patients without prior treatment and low-risk SMM patients (without progression to primary myeloma) may be informative for identifying features that could predict high-risk versus low-risk SMM.*

Reviewer 1: Page 8 – Correlation of clustering by translocation. Are there sufficient samples to claim that this is a broad phenomenon? If so, what is the relevance?

Authors: *As with a few other comments, this one touches upon sample size. Multiple myeloma is especially tricky because even its most frequent driver events are only found in small percentages of patients, necessitating very large samples for drawing general conclusions. While we cannot make such conclusions here, it is important to investigate the assumption that “samples with similar cytogenetic alterations will tend to have similar gene expression profiles”. Indeed, 77570 and 83942 both have t(11;14) and exhibit similar expression profiles, which provisionally supports this assumption. If it is later found to hold true in a larger sense, it should be possible to further investigate the features for the associated expression profile, which could shed light on additional therapeutic targets for a certain group of patients.*

Reviewer 1: Page 8 – Plasma cells cluster by patient, non-PC by cell type. This is hardly surprising. Of the two SMM samples, there appear to be very few PC in 58408. Is this correct?

Authors: *The observation is consistent with the expectation that tumor cells are highly heterogeneous, while the microenvironment tends to be more similar across patients. But, it is worth investigating the degree to which this diversity is functionally significant for disease physiology. We thus utilized scRNA-seq to examine drivers of cluster resolution. For the SMM time point of 58408, we indeed did not find a lot of plasma cells in the sample.*

Reviewer 1: Page 8 – Differential expression analyses. Conclusions are drawn from only a few cases: two SMM and one remission. Difficult to draw any conclusions here. Relevance? There are six relapse samples, but only three (for IL1R2) or one (for IL1B) are shown in the figure. Why is this? What about the other relapse samples? Does this in fact indicate that the phenomena are not general? Why do the findings matter?

Authors: We the referee’s comment regarding consistency of microenvironment profiles across samples, but unfortunately, the observation we reported in the manuscript is not seen across all samples (Reviewer Response Letter Figure 2). For cases 27522 and 56203, IL1R2 expression in monocytes is not altered significantly across time points. For IL1B, only 60359 showed an obvious change. This suggests the findings may not be general, given the diverse spectrum of patient genomic landscapes and treatment regimes. Nevertheless, we were able to use our longitudinal sampling to pinpoint some interesting microenvironmental changes at the cell-type level. It would take a significantly greater number of samples to support any generalizations regarding the microenvironment that would be helpful for guiding future therapeutic interventions.

Response Letter Figure 2. Expression of IL1R2 and IL1B in monocytes for samples with multiple time points.

Reviewer 1: Page 8/9 – Three cases examined for microenvironmental cells. There is no consistent pattern. What is the biological relevance?

Authors: Our cohort is a heterogeneous collection that represents many genomic variations, disease stages, and treatment regimes, so inconsistencies in the microenvironment are not wholly unexpected. However, due to the large body of work detailing the interaction and interdependence of myeloma with other cells in the bone marrow, it may be surprising that we do not see strong indications of specific microenvironmental perturbations. We believe that reporting this observed heterogeneity is therefore important in underlining the complexity

of this disease. Future studies with a more homogenous cohort may better elucidate the role of the microenvironment in myeloma.

Reviewer 1: Page 9 – Outlier analysis on two patients. The same problem. Can we draw general conclusions about t(11;14)? What is the relevance?

Authors: *Here, we test the assumption that “tumors with similar cytogenetic alterations may have similar microenvironment profiles”. While multiple myeloma is known to be highly interactive with its physiological niche, the correlation between tumor subtype and specific microenvironment perturbations has not yet been heavily studied. We explore this question using scRNA-seq of whole bone marrow, which precludes biases due to cell-sorting that hinder other methods. While our cohort is admittedly too small for general conclusions, we feel that this study provides valuable evidence for a presently sparse body of work.*

Reviewer 1: Page 10/11 – B-cell lineage analysis. A purported association between MS4A1 expression in PCs of patients 56203, 83942, 77570 and CCND1 regulation. What about patient 81012, which has a t(11;14) but very low MS4A1? Why doesn't this show the pattern?

Authors: *This observation serves to validate the proposed molecular subtype classification of MM by Zhan et al.¹⁶. In that study, patients with CCND1 activation were classified into two groups, where one group, CD2, is characterized by high expression of certain B cell markers like MS4A5 (CD20) and PAX5, while the other group, CD1, lacks B cell markers and exhibits overexpression for genes FYN and SETD7. For patient 77570 and 83942, we observed high expression of MS4A1, suggesting those two cases belong to the CD2 group. Conversely, case 81012 showed elevated SETD7 and FYN expression, suggesting it belongs to the CD1 group.*

Reviewer 1: What is meant by “aberrant CCND1 regulation according to FISH”? It is not a translocation and there is no overexpression, so what are we talking about?

Authors: *Unfortunately, our explanation regarding this point was somewhat ambiguous. The statement mentioning “aberrant CCND1 regulation according to FISH” actually refers to the results from the FISH report. FISH was not able to detect the IGH-CCND1 rearrangement for case 56203; however, an abnormal pattern consisting of three signals for IGH and two signals for CCND1 was observed in 87/200 nuclei, indicative of trisomy for the corresponding region on chromosome 14.*

¹⁶ Zhan, F. et al. The molecular classification of multiple myeloma. Blood 108, 2020-2028 (2006).

Reviewer 1: The authors really cannot draw any conclusion about relationship between CCND1 and B-cell phenotype from their data. Even if they could, they still fail to explain why this is relevant. If they were speculating about pre-germinal centre origins of myeloma or cells that manifest resistance, this might be interesting, but I cannot see specific evidence of this in the data that they show.

Authors: *As mentioned, this observation supports a previous study focusing on different subtypes of multiple myeloma. Based on single cell data alone, however, the only conclusion here is that some myeloma cells with aberrant CCND1 regulation exhibit a B cell phenotype, suggesting targeting B cells could be another effective myeloma treatment regime. The underlying reason for the observed B cell phenotype is not clear, but it could be that some plasma cells de-differentiate into a less mature state, or that there are malignant cells that are already lurking at premature stages. It would be definitely interesting to investigate whether the resistant myeloma cells are truly of pre-germinal center origins, but it is beyond the scope of our current study.*

Reviewer 1: Page 11 – Having been shown four levels of B-cells, we are not told why this is important.

Authors: *We appreciate the reviewer's raising this point. We have added an explanation to the revision, as follows.*

- 1. By dissecting lineage-specific genes, we can discern whether malignant plasma cells exhibit high expression of certain genes that are indicative of earlier lineages (mature B cells, earlier/primitive B cells).*
- 2. It allows highlighting of features that distinguish between normal and malignant plasma cells at the single cell level.*
- 3. It helps identify novel and less-explored markers for each developmental stage that might be worth further investigation.*
- 4. Our observations could serve as a reference for other scRNA-seq studies for cross-comparison, especially in the field of myeloma and other B cell malignancies.*

Reviewer 1: Page 11/12 – Deep chromosome 13 loss in three patients. From the manuscript: “Clusters with deeper deletion tended to be patient and subpopulation-specific, while cells mapping to the same location as normal plasma cells tended to come from multiple patients and showed greater variability”. This is very vague. It is not formalized or quantified. Why does it matter?

Authors: *To address the referee's point, we have added a mapping for normal plasma cells and changed the layout of color for better and clearer visualization. A figure is also attached below (reviewer response letter figure 3) to show the distribution of B cells and plasma cells (with copy number information for chromosome 13) for each sample. It should be noted, though, that due to the limited number of plasma cells from healthy donor samples, these cells exhibit a scattered pattern in the tSNE (but still within the same region). This observation is important because the patient and subpopulation-specific pattern for clusters with deeper*

chr13 deletion suggests that deep deletion of chr13 is an important feature in determining the overall expression profile of malignant plasma cells. Conversely, if this feature is not important, we would expect that samples with and without deep chr13 loss would be mixed together. Taking a step further, deep chr13 loss could be a stratifying feature for helping determine treatment regimes in the clinic (assuming there are enough samples for a solid conclusion). We have revised the text in the manuscript to clarify this point.

Response Letter Figure 3. Mapping of B cells and plasma cells (with copy number information for chromosome 13) to the integrated tSNE plot with B and Plasma cells for each sample, respectively.

Reviewer 1: Page 12 – Investigation of clonal structure. Only three samples. Generalizable? Relevant?

Authors: *The reviewer raises the questions of generality and relevance of the manuscript's clonal structure analysis. All three cases are collected at SMM and primary diagnosis stages, with the consistency in timing for all indicating this structure is characteristic of at least some fraction of cases that progress from SMM to MM. A very simple probability model that posits picking 3 such samples randomly from a population would be significant at the 0.01 level suggests a population fraction around $0.01^{1/3} \approx 21\%$. We certainly concede that this observation might not be the only existing situation, but we do maintain this observation bears at least partial generalizability. Its relevance includes the following:*

- 1. Transition from SMM to MM does not necessarily involve significant, dynamic cell subpopulation changes (which is already suggested in the manuscript).*
- 2. While not mentioned in the manuscript, a further indication for this observation is that there could be features within plasma cells that distinguish high-risk SMM (which would progress to MM) from low-risk SMM (which remains stable and does not progress). Identifying features that distinguish high/low-risk SMM is a completely different undertaking that requires a different experimental design though.*

Reviewer 1: Page 14 – The authors describe two t-SNE clusters in each of primary and relapse samples of six patients. Here, at last, we might have numbers that allow us to generalize a finding. However, we should also presumably see a similar two clusters in all the other primary samples if this is a general phenomenon – I don't think this is shown. Assuming that this is the case, what is the situation in SMM? Is there just one cluster in SMM then two in primary myeloma or are there always two clusters? Ultimately, we are still left with the question: why does this matter for myeloma biology?

Authors: *Indeed, we did not find a similar two-cluster-structure for all primary samples, which is also reflected in Figure 7a, but which might have been expected due to the specificity of the malignant plasma cells from different patients. Our analysis suggests the transition from SMM to primary MM should be a process in which plasma cell populations remain generally stable; therefore, we would assume the population structure for the cases where we did not find two-cluster-structure remains the same for its corresponding SMM time point. However, due to the unavailability of such samples, we do not have sufficient evidence to make such predictions in the manuscript. Rather, we are expecting future studies would be able to investigate a larger number of cases where SMM and primary diagnosed longitudinal samples are available for a more comprehensive characterization, in order to make more solid conclusions.*

For the cases we showed regarding the two-cluster structure, we discussed possible biological interpretations in later sections. Briefly, the two clusters are usually characterized by different levels of FOS/JUN expression and we were able to find downstream targets by

examining FOS binding sites, suggesting ZBTB20 and H3F3B are two downstream targets that are being regulated. Analysis for the presumed downstream targets of ZBTB20 and H3F3B points to a senescent phenotype for cells with high expression of ZBTB20 and H3F3B, characterized by presence of SAHF, enhanced survival, decreased proliferation, and increased inflammatory cytokines. Altogether, the evidence suggests the AP-1-high population within the two-cluster structured plasma cells is associated with senescence. For samples from more advanced disease stages, though (e.g. Relapse), there are situations where AP-1-high population exhibit different biological meanings, which are discussed in the manuscript and the following response letter as well (e.g. page 14 about 81012, page 17 about 56203)

Reviewer 1: Page 14/15 – Proportion of PCs varies between primary and relapse (but not always?). Both increases and decreases are seen. What does this tell us?

Authors: *PC proportion may reflect tumor burden, but it may also be due to sampling variation. While we report observed proportions, we stress that bone marrow sampling is inherently noisy, and we suggest that PC proportion not be viewed as a direct indicator of tumor burden.*

Reviewer 1: Page 15 – Patient 81012 subpopulation dynamics.

Authors: *The reviewer's concern for 81012 subpopulation dynamics is well taken. We have splitted our response to concerns into pieces in order to address them individually in the following pages.*

Reviewer 1: “Trajectory-based analysis also suggests R.3 and R.4 are mainly present at the end point of state 4 and state 5, respectively.” What exactly are states 4 and 5, apart from some arbitrary point in pseudotime? Why is this important?

Authors: *A “state” in pseudotime analysis lies between two adjacent branch points, or between a branch point and a terminal point. We performed trajectory analysis using Monocle (R package), whose algorithm assumes that cells undergo traceable, step changes in phenotype, and that a sample consists of cells along different stages of this developmental progression. Monocle infers how cells align along this pseudotime trajectory based on degrees of similarity in their gene expressions. States 4 and 5 are two terminal states - the cells at their respective terminal points are furthest away from the rest of plasma cells and B cells. We found R.3 and R.4 to be present mainly at the end points of states 4 and 5, which suggests that these two populations have different expression profiles from other cells. This is just one piece of the larger body of evidence showing that R.3 and R.4 are two distinct, new populations.*

Reviewer 1: More confusingly, we are told first that P.2/R.2 have the lowest expression of FOS. We are then told that P.1, R.1, and R.2 exhibit higher expression. So does R.2 exhibit higher or lower expression?

Authors: *Based on figure 4f, P.2 and R.2 are the populations with lower FOS expressions. The previous text was due to a typo and we apologize for the confusion. The corrected sentence should be: "For FOS, one component within the AP-1 complex, we found the lowest expression in P.2 and R.2; P.1, R.1 and R.3 exhibit higher expression, while R.4 shows highest expression." We very much appreciate the reviewer's diligence in catching this mistake and we have corrected the sentence in the manuscript.*

Reviewer 1: Similarly, R.4 is stated to have the highest expression, but we are then told that R.3 and R.4 have similar expressions of FOS to R.1. R.4 has chr 19 loss, but we are not told why this matters (chromosome 19 loss is not one of the more common cytogenetic findings in myeloma).

Authors: *The reason we specifically looked into R.3 and R.4 is that both populations are newly-derived, indicating the differences for these two populations between R.1/R.2 is not only associated with FOS expression, although both populations have high FOS expression (despite even higher in R.4). We again appreciate reviewer's diligence and have modified the text in the manuscript to clarify our rationale. We agree that chromosome 19 loss is not a common cytogenetic finding in myeloma. However, the reason for including inferCNV analysis here is that we wanted to see if there is any evidence at the CNV level that could distinguish different myeloma populations. It turned out that, among all the global CNV alterations, chr 19 loss within this case is most striking. For R.4, we think chromosome 19 loss is another layer of evidence, besides expression and geometric location within UMAP, to demonstrate that this is a newly-derived, distinct population and that this population is different from others in terms of gene expression and copy number profile.*

Reviewer 1: The conclusion is that R.3 and R.4 have enhanced growth and survival, but I don't see what data tell us that. Are they basing this on overexpression of CKS1B and MEF2C, respectively? If so, this is a bizarre conclusion. Surely there are multiple other factors that can promote growth and survival that may be expressed slightly higher and/or in combination by P.1 and P.2 (but not making the threshold for differential expression). Again, even if true, why does it matter? Can we say that it is a universal/near-universal phenomenon?

Authors: *The reviewer indicates that analyzing the overexpression of individual genes is not sufficient to support our conclusion. Accordingly, we examined across all top DEGs for R.3 and R.4, respectively (Reviewer Response Letter Figure 4a; shown are the top 15 DE genes for R.3 and R.4, respectively).*

In fact, CKS1B is not among the top members in the list of DE genes for R.3; however, we do find genes such as MKI67 and TOP2A, two common proliferative markers, to be among the top list. Based on results from Reactome-based pathway analysis, the top DE genes for R.3 are enriched for pathways such as cell cycle, G1/S-Specific Transcription (Reviewer Response Letter Figure 4b), providing another layer of evidence for enhanced growth of R.3.

For R.4, we are not able to find DE genes obviously enriched in specific pathways. This is partially expected, though, given the presence of lncRNAs (MALAT1, NEAT1, SMCR5) and that some of the genes are less well-explored (CTB-152G17.6, AC104532.4). Nevertheless, we are able to find upregulation of CCND1 for R.4, apart from MEF2C alone. Also, we also find upregulation of ADAR, an RNA-editing enzyme that has recently been reported to promote MM progression¹⁷.

Altogether, we show that, R.3 and R.4, the two distinct populations, exhibit unique expression profiles than all the other populations and that such expression profiles are associated with altered biological processes that could contribute to disease progression through a variety of mechanisms.

Response Letter Figure 4. Additional analysis for dynamics within case 81012.

(a) Scaled expression for top DE genes from R.3 and R.4, respectively. Colors indicated average of scaled expression; size of the dots indicates the percentage of cells with non-zero expression. (b) Pathway enrichment analysis for top DE genes from R.3.

The referee's question regarding the universality of this phenomenon cannot really be answered meaningfully at this time because of the complexity of the disease. Nevertheless, this is a very interesting question and we expect it will be addressed in the future with a more uniform sample collection strategy.

Reviewer 1: Page 16 – Similar analysis for 56203.

Authors: Similar to case 81012, the reviewer is concerned about the biological interpretation for case 56203, and we will address the questions point by point in the following responses.

¹⁷ Tasakis, R. N. et al. ADAR1 can drive Multiple Myeloma progression by acting both as an RNA editor of specific transcripts and as a DNA mutator of their cognate genes. bioRxiv (2020).

Reviewer 1: (1) They claim that P.1 is closest to the relapse samples, but extended data figure 8a is really hard to follow and it is not clear how they concluded this.

Authors: *The difficulty mentioned by the reviewer may be due to the multiple-color regimes being used; for better illustration, we have attached another figure below (Reviewer Response Letter Figure 5a and 5b).*

For the initial round of plasma cell subcluster analysis, we were able to find 3 subclusters from primary (P.1, P.2, P.3) and 2 subclusters from relapse-1 (R.1, R.2). We further divided P.1 into 6 “sub”-subclusters using the “FindClusters” function from R package Seurat; this 2-nd level subcluster is created in the hope of finding a more refined mapping pattern, given the large number of cells from P.1. Therefore, in the extended data figure 8a, there are multiple colors representing different subclusters.

In Reviewer Response Letter Figure 5b, we are showing the initial subcluster mapping of 56203. We can observe clusters R.1 (green) and R.2 (dark orange) cluster separately from P.1 (pink), P.2 (purple), and P.3 (yellow) in the integrated tSNE plot. However, based on geometric distribution of the cells, R.1 and R.2 are still closest to P.1, but are distant from P.2 and P.3. This information, together with other evidence from the expression level (e.g. the expression for SDC1, SLAMF7 and CCNL1 are higher for P.1, R.1 and R.2 and lower for P.2 and P.3), leads us to speculate that R.1 and R.2 are derived from P.1, while the other two populations from primary stages are lost as disease progresses, as shown in the bottom left of extended figure 8a.

Reviewer 1: (2) Assuming, though, that they are correct in this, how many cells exhibited chr 17 loss in P.1. Were they present before treatment?

Authors: *We checked chr17 loss based on averaged scaled chr17 copy number profiles and used 0.76 as the cutoff for copy number loss. Within P.1, we found 34/2015 cells with chr17 copy number loss. For P.2 and P.3, we found 0 cells with chr17 loss out of 297, 255 cells, respectively. For R.1 and R.2, we found 96/131, 27/32 cells with copy number loss, respectively. While we were not able to find a huge proportion of cells with chr17 loss before treatment for P.1, this is the only population from the primary stage where we could observe copy number loss. It could suggest some of the cells from P.1, rather than P.2 or P.3, gained growth advantage, and likely additional phenotypic changes, along treatment, and became the resistant populations found for R.2 and R.3.*

Reviewer 1: (3) Surely p53 is the best candidate for a tumor suppressor on chr 17. Why are we not shown p53 expression in the various samples?

Authors: *The reviewer is suggesting that we check the expression of TP53 as a representative for chromosome 17 loss. However, we did not really expect obvious gene expression changes from the scRNA side (Reviewer Response Letter Figure 5f). This is actually due to the sparse nature of single cell RNA-Seq data. In scRNA analysis, sparsity means the observation where a given gene within a given cell has no unique molecules*

mapped to, which could be either true absence of expression, or due to drop out (the gene is expressed, but not detected by the platform). As expected, TP53 exhibits low expression (with a large proportion of zero-reads) across different subclusters; as a result, it would be difficult to infer chromosome 17 loss from TP53 expression.

The rationale for using inferCNV as a measurement for copy number profile is that, copy number profile could be proximal to the average gene expression for large genomic regions; the caveat for drop out within a specific gene could be compensated for, given a larger number of genes present in a certain genomic region¹⁸. Therefore, we believe the output from inferCNV should be a better representation of the copy number profile.

Reviewer 1: (4) And, again, what is the biological relevance of all this?

Authors: The take-away here is that we can detect dynamic tumor population shifts during disease progression: in 81012, R.3 and R.4 are newly gained groups relative to primary tumor populations, while in 56203, P.2 and P.3 are lost at relapse. These two cases contrast with 58408, which exemplifies population stability from SMM to the MM stages.

For 56203, one hypothesis is that P.2 and P.3 are eliminated after treatment, while P.1 represents a refractory subclone. Since R.1 and R.2 do not overlap exactly with P.1 cells in the integrated tSNE, we see that these additional relapse subclones have gained features not present in P.1. The features that set R.1 and R.2 apart from P.1 are shown in the dot plot below (Reviewer Response Letter Figure 5c). Among these features, the ones gained in R.1 (relative to P.1) are generally shared by R.2, whereas some features are unique to R.2.

Pathway enrichment analysis for R.1 features highlight the Wnt and cytokine signaling (Reviewer Response Letter Figure 5d); in contrast, R.2 features enrich for cell cycle pathways (Reviewer Response Letter Figure 5e). This suggests that R.2 in 56203 exhibits enhanced proliferation. Interestingly, it appears similar to R.3 in 81012, a newly gained tumor population in relapse that upregulates TOP2A, MKI67, and CKS1B. Although we have too few samples for further generalization, we show quantifiable changes in tumor subclonality. Our results represent a promising line for future investigation into tracing the origin of a highly proliferative clone in relapse stages.

¹⁸ inferCNV of the Trinity CTAT Project. <https://github.com/broadinstitute/inferCNV>

Response Letter Figure 5. Additional analysis for dynamics within case 56203.

(a) Duplication of Extended Data Figure 8a. (b) Mapping of subcluster to tSNE plot for the integration of primary and relapse-1 from case 56203. (c) Dot plot showing the scaled expression for the genes differentially expressed for R.1 or R.2. (d-e) Pathway enrichment analysis showing the top-enriched pathways from top DE genes from R1 (d) and R.2 (e); (f)

Predicted copy number profile of chromosome 17 (upper) and expression of TP53 (bottom) for each subpopulation within case 56203.

Reviewer 1: Page 16/17 – The analysis of patient 27522 is poorly written and hard to follow. They conclude by stating that there are “important clinical implications” but don’t tell us what these are. The statement: “Taken together, one interpretation is there were cells present at remission that evaded treatment and survived to seed the relapse” seems to be a remarkably anodyne conclusion; any clinician who has treated myeloma knows this already.

Authors: *In relapse-2 of patient 27522, we found three plasma cell subpopulations with different degrees of predicted malignancy based on their distinct mutation, gene copy-number, and gene expression profiles. Technical limitations, along with a very low number of detectable plasma cells at the remission stage, limit our ability to identify the exact clone that may have given rise to the relapse subpopulations. Nevertheless, we established evidence at the single cell level that even at relapse, a spectrum of phenotypes exist within the tumor. As the reviewer points out, current disease models are well familiar with subclonal expansion driven by selective pressures of myeloma treatment -- here, we add to this understanding by identifying the specific genomic and transcriptomic composition of this presupposed tumor diversity. We believe that this is valuable, albeit limited, insight into how single cell techniques have the potential to inform and enhance future treatment strategies.*

Reviewer 1: Page 21 – It is not clear how fig. 7A supports the text, which is not easy to follow. Are the authors drawing their conclusions on the role of AP-1 in myeloma from fig. 7A (which does not support the conclusion) or from samples 58408 and 81012, which is only two patients?

Authors: *The reviewer is wondering about the rationale for drawing conclusions from Fig 7A, where we stated that “We then evaluated the expression of FOS and JUN across subclusters and across samples, finding at least one plasma cell subpopulation with high expression of FOS or JUN in all cases, regardless of AP-1 expression differences (Fig. 7a).” We concede that there is some ambiguity in the writing.*

We did not draw our conclusion from 58408 or 81012, however, these are the two typical cases that prompt us to investigate the subcluster structures of plasma cells across samples with a specific focus on AP-1 complex components FOS and JUN. For the subcluster analysis, we looked into the tSNE plot for plasma cell distribution for each individual samples and manually gated subpopulations based on the geometric locations in tSNE plot, when possible (the strategy is similar to what we did for 58408 and 81012 in Figure 4a and 4d; specific tSNE plots not shown due to space limit). It turns out that for each sample, whether there is only one subcluster or more than one subcluster, at least one of the subclusters exhibits a high expression of either FOS or JUN when comparing against the plasma subclusters across the cohorts (7A, upper panel heatmap). The bottom panel for the violin plot shows the expression for certain cases at a more zoomed-in perspective. It can be seen that for both 37692 and 47491, both S.2 and P.1 exhibits relatively elevated FOS and JUN

expression; for 56203, P.1, R.1 and R.2 all show high expression; for 57075, the expression level of FOS and JUN between the two subclusters is not readily distinguished, but both exhibit high expression.

We have edited the manuscript text to remove the statement “Interestingly, plasma cells from the multiple sample collections of Patients 58408 and 81012 showed subpopulations exhibiting differential expression of both FOS and JUN, and we manually defined plasma cell subclusters for each sample based on their t-SNE mapping location” to minimize the confusion.

Reviewer 1: Fig. 7B shows some correlation between FOS/JUN expression and a few of their known targets (IL6ST and IL1A). This is entirely unsurprising, but doesn't really tell us anything.

Authors: *We appreciate the reviewer's attention to this detail. We included CyTOF protein quantitation results in Fig 7B and 7C to validate the transcriptomic expression patterns that we see in scRNA analysis. The differences in FOS/JUN expression across plasma cell subpopulations suggest that our in-silico findings are true at the protein level, as well. We first verified the interaction between FOS/JUN and IL6ST using ChIP-Seq analysis designed to find FOS-interacting promoter regions in B-cell lymphoma. While the interaction between FOS/JUN and these targets is not an entirely novel finding, the direction of regulation, especially in the context of myeloma, is not well-elucidated in present literature to the best of our knowledge. The physiological consequences of FOS/JUN upregulation in this disease are not well understood; we therefore aim to provide a closer look at downstream targets by analyzing correlation with putative interactors. We have refined the text for clarity.*

Reviewer 1: Fig. 7C This shows a possible correlation between FOS and H3F3B for only four samples and no or minimal correlation between FOS or JUN and other potential targets. The CyTOF data appear to be mostly missing. I am not sure what we can conclude from this. In terms of the bottom of the figure (i.e. the functional consequences), this is speculative. Many of the proteins may be pleiotropic and, as noted, many of the correlations appear weak. Surely some of these proposed functional consequences are potentially testable, e.g. are there any gene signatures or expression markers of SAHF that could be examined (with the caveat that we are only looking at four samples).

Authors: *We acknowledge the reviewer's concern about the validity of the experimental design. CyTOF experiments here mainly serve as a method of validation; besides, the number of cells required for one run of CyTOF exceeds the number of cells needed for a single cell experiment, which already limits our choices of samples. Therefore, only four samples were included in the CyTOF experiment.*

The correlation between FOS/JUN and other potential targets seems to be minimum or does not exist at all based on the visualization. This is likely due to the low expression baseline and potentially high drop-out rates for some of the genes (regarding scRNA data). Therefore, the average expression for genes such as ZBTB20 and IL6ST seem to be very low, with the

color looking similar for the AP-1 high/low populations; but the trend for AP-1 complex expression and the potential targets are consistent. A similar scenario applies for protein data as well, where the high expression of JUN at the protein level partially masks the expression patterns for other potential downstream targets. In terms of CyTOF data being missing, this is due to the unavailability of antibodies for some of the genes of interest.

SAHF stands for senescence-associated heterochromatin foci, which is one of the markers for senescence. Experimental validation of SAHF by itself requires staining¹⁹, which is not included in our study design. Additional ways to go around with this question is the examination of the senescence phenotype by itself. In fact, our conclusion for the AP-1 high population in general, is that this is a population with associated with senescent phenotype, and the evidences, apart from SAHF, also includes enhanced survival (e.g. MCL1), decreased cell proliferation (e.g. CDKN1A), and presence of senescence-associated-secretory profile (IL1B).

Reviewer #2 (Remarks to the Author): Expert in single-cell RNA-seq, biostatistics, and bioinformatics

Reviewer 2: The authors have addressed my concerns from the last round of reviews and I have no more reservations. As I wrote last time, this study is done on a small cohort that is very heterogeneous, and most of the results obtained are inconclusive. But, I appreciate the reviewers' point that multiple myeloma is an extremely heterogeneous disease, and it is not realistic right now to conduct a study of the scale necessary to draw integrative general conclusions. I think this is a nice demonstrative study, with successful integration across multiple omics data types, that provides a data resource and initial exploration upon future large scale studies can be based.

Authors: *We appreciate the referee's observations regarding the heterogeneity of sampling during the last round of revision. We would envision our study to serve as a foundation for larger scale studies in the future.*

Reviewer #3 (Remarks to the Author): Expert in single-cell RNA-seq and tumour microenvironment

Reviewer 3: The authors have addressed all of my comments in great detail. They have thoroughly reanalysed the single-cell RNA-seq data and included many new analyses of the tumour microenvironment as well as trajectory analysis of the plasma cells, which is a tremendous effort. However, even though the study provides a unique longitudinal dataset and represents a valuable resource, the more detailed analyses still haven't resulted into common trends and significant insights about the progression or treatment of myeloma. The samples are very heterogeneous in terms of their mutational and transcriptional landscape. The manuscript might be more suitable to be published as a resource.

¹⁹ Aird, K. M. & Zhang, R. Detection of senescence-associated heterochromatin foci (SAHF). *Methods Mol Biol* 965, 185-196 (2013).

Authors: *We would like to thank the reviewer for providing constructive suggestions within the revision. As the reviewer pointed out, that our sample size is not sufficient to conclude common trends and significant insights into the myeloma disease progression, we agree that being published as a resource may be more suitable.*

References

1. Bezieau, S. et al. High incidence of N and K-Ras activating mutations in multiple myeloma and primary plasma cell leukemia at diagnosis. *Hum Mutat* 18, 212-224 (2001).
2. Efremova, M., Vento-Tormo, M., Teichmann, S. A. & Vento-Tormo, R. CellPhoneDB: inferring cell-cell communication from combined expression of multi-subunit ligand-receptor complexes. *Nat Protoc* 15, 1484-1506 (2020).
3. Neri, P. et al. Neutralizing B-cell activating factor antibody improves survival and inhibits osteoclastogenesis in a severe combined immunodeficient human multiple myeloma model. *Clin Cancer Res* 13, 5903-5909 (2007).
4. Chen, J. et al. BAFF is involved in macrophage-induced bortezomib resistance in myeloma. *Cell Death Dis* 8, e3161 (2017).
5. Yu, B., Jiang, T. & Liu, D. BCMA-targeted immunotherapy for multiple myeloma. *J Hematol Oncol* 13, 125 (2020).
6. Smith, E. L. et al. GPRC5D is a target for the immunotherapy of multiple myeloma with rationally designed CAR T cells. *Sci Transl Med* 11, eaau7746 (2019).
7. Orchard, S. et al. Protein interaction data curation: the International Molecular Exchange (IMEx) consortium. *Nat Methods* 9, 345-350 (2012).
8. Luck, K. et al. A reference map of the human binary protein interactome. *Nature* 580, 402-408 (2020).
9. Bystry, R. S., Aluvihare, V., Welch, K. A., Kallikourdis, M. & Betz, A. G. B cells and professional APCs recruit regulatory T cells via CCL4. *Nat Immunol* 2, 1126-1132 (2001).
10. Abe, M. et al. Role for macrophage inflammatory protein (MIP)-1alpha and MIP-1beta in the development of osteolytic lesions in multiple myeloma. *Blood* 100, 2195-2202 (2002).
11. Reiners, K. S. et al. Soluble ligands for NK cell receptors promote evasion of chronic lymphocytic leukemia cells from NK cell anti-tumor activity. *Blood* 121, 3658-3665 (2013).
12. Pittari, G. et al. Restoring Natural Killer Cell Immunity against Multiple Myeloma in the Era of New Drugs. *Front Immunol* 8, 1444 (2017).
13. Kuttruff, S. et al. NKp80 defines and stimulates a reactive subset of CD8 T cells. *Blood* 113, 358-369 (2009).
14. Tong, B. & Wang, M. CD47 is a novel potent immunotherapy target in human malignancies: current studies and future promises. *Future Oncol* 14, 2179-2188 (2018).
15. Fan, F. et al. The AP-1 transcription factor JunB is essential for multiple myeloma cell proliferation and drug resistance in the bone marrow microenvironment. *Leukemia* 31, 1570-1581 (2017).
16. Zhan, F. et al. The molecular classification of multiple myeloma. *Blood* 108, 2020-2028 (2006).
17. Tasakis, R. N. et al. ADAR1 can drive Multiple Myeloma progression by acting both as an RNA editor of specific transcripts and as a DNA mutator of their cognate genes. *bioRxiv* (2020).
18. inferCNV of the Trinity CTAT Project. <https://github.com/broadinstitute/inferCNV>
19. Aird, K. M. & Zhang, R. Detection of senescence-associated heterochromatin foci (SAHF). *Methods Mol Biol* 965, 185-196 (2013).

REVIEWERS' COMMENTS

Reviewer #1 (Remarks to the Author):

I appreciate the authors' considerable work in addressing my criticisms. Their point about the difficulties of achieving sufficient power to make substantive conclusions is well-made.

I also note that the other reviewers were happy with the manuscript on first submission.

My only remaining criticism is that the SRA submission of their data has not yet been completed. This is an important dataset and making the data publicly available would be of great value to the cancer research community. I would hope that the submission is completed prior to publication.

REVIEWERS' COMMENTS

Reviewer #1 (Remarks to the Author):

I appreciate the authors' considerable work in addressing my criticisms. Their point about the difficulties of achieving sufficient power to make substantive conclusions is well-made.

I also note that the other reviewers were happy with the manuscript on first submission.

My only remaining criticism is that the SRA submission of their data has not yet been completed. This is an important dataset and making the data publicly available would be of great value to the cancer research community. I would hope that the submission is completed prior to publication.

Authors:

We appreciate the reviewer's careful going through our manuscript and raising constructive suggestions. Regarding the reviewer's last concern about data availability, we have uploaded the raw sequencing data to SRA preload folder and waiting for the SRA to complete with submission processing [<https://submit.ncbi.nlm.nih.gov/subs/sra/SUB8614413/overview>]. Meanwhile, data availability statement is updated in our manuscript.